# Drug-like sphingolipid SH-BC-893 opposes ceramide-induced mitochondrial fission and corrects diet-induced obesity

Vaishali Jayashankar[1],[†], Elizabeth Selwan[1],[†], Sarah E Hancock[2], Amandine Verlande[3], Maggie O Goodson[3], Kazumi H Eckenstein[1], Giedre Milinkeviciute[4], Brianna M Hoover[5], Bin Chen[6], Angela G Fleischman[5], Karina S Cramer[4], Stephen Hanessian[6], Selma Masri[3], Nigel Turner[2] & Aimee L Edinger[1],*

## Abstract

Ceramide-induced mitochondrial fission drives high-fat diet (HFD)-induced obesity. However, molecules targeting mitochondrial dynamics have shown limited benefits in murine obesity models. Here, we reveal that these compounds are either unable to block ceramide-induced mitochondrial fission or require extended incubation periods to be effective. In contrast, targeting endolysosomal trafficking events important for mitochondrial fission rapidly and robustly prevented ceramide-induced disruptions in mitochondrial form and function. By simultaneously inhibiting ARF6- and PIKfyve-dependent trafficking events, the synthetic sphingolipid SH-BC-893 blocked palmitate- and ceramide-induced mitochondrial fission, preserved mitochondrial function, and prevented ER stress in vitro. Similar benefits were observed in the tissues of HFD-fed mice. Within 4 h of oral administration, SH-BC-893 normalized mitochondrial morphology in the livers and brains of HFD-fed mice, improved mitochondrial function in white adipose tissue, and corrected aberrant plasma leptin and adiponectin levels. As an interventional agent, SH-BC-893 restored normal body weight, glucose disposal, and hepatic lipid levels in mice consuming a HFD. In sum, the sphingolipid analog SH-BC-893 robustly and acutely blocks ceramide-induced mitochondrial dysfunction, correcting diet-induced obesity and its metabolic sequelae.

**Keywords** ceramide; high-fat diet; leptin resistance; mitochondrial fission; obesity
**Subject Categories** Metabolism; Pharmacology & Drug Discovery

See also: **C Muley & A Bartelt** (August 2021)

## Introduction

Obesity has emerged as a serious epidemic. According to the World Health Organization, an estimated 13% of the world's adult population was obese in 2016, a number that has nearly tripled since 1975. Even more alarming is the accelerating prevalence of obesity in children. Worldwide, over 340 million children and adolescents aged 5–19 were overweight or obese in 2016; most of these individuals will eventually become obese adults. Because obesity and related comorbidities are leading causes of preventable and pre-mature death (Abdelaal et al, 2017), these statistics reflect a staggering social and economic burden. While the drivers of the growing obesity epidemic are multi-factorial, over-consumption of calorie dense, high-fat foods clearly contributes (Swinburn et al, 2011). These hypercaloric diets synergize with environmental and genetic factors to create a chronic positive energy balance that leads to excessive adiposity and metabolic dysfunction. While dietary modification and increasing exercise are an integral part of any interventional program, lifestyle changes alone have proven insufficient to resolve obesity in most patients. The "eat less and move more" approach ignores the complex physiologic, psychological, and genetic factors that prevent some patients from maintaining a negative energy balance (Roberto et al, 2015). While bariatric surgery can be highly effective, it is also invasive and can be accompanied by serious complications (Nguyen & Varela, 2017). Thus, there is a critical unmet need for medical therapies that can complement dietary and lifestyle interventions and help individuals overcome the barriers to successful long-term weight loss.

1 Department of Developmental and Cell Biology, University of California Irvine, Irvine, CA, USA
2 School of Medical Sciences, University of New South Wales, Sydney, NSW, Australia
3 Department of Biological Chemistry, University of California Irvine, Irvine, CA, USA
4 Department of Neurobiology and Behavior, University of California Irvine, Irvine, CA, USA
5 Division of Hematology/Oncology, Department of Medicine, University of California, Irvine, CA, USA
6 Department of Chemistry, Université de Montréal, Montréal, QC, Canada
*Corresponding author. Tel: +1 949 824 1921; E-mail: aedinger@uci.edu
†These authors contributed equally to this work

FDA-approved weight-loss agents are only marginally effective and often plagued by toxicities and side effects (Bessesen & Van Gaal, 2018; Srivastava & Apovian, 2018). Our growing understanding of the signals that control satiety and metabolism, especially the hormonal crosstalk between peripheral tissues and complex neural circuitry, has identified several new targets that may be more amenable to pharmacological intervention (Williams *et al*, 2020). The discovery of the hormone leptin almost 3 decades ago was particularly exciting, offering hope that obesity could be treated by modulating leptin signaling (Andreoli *et al*, 2019). Although pharmacological administration of leptin dramatically decreases food intake and increases energy expenditure in the rare patients with leptin deficiency, leptin itself has limited potential as an obesity therapeutic. The majority of individuals with obesity are leptin resistant, a phenotype that has been traced to both hyperleptinemia (Zhao *et al*, 2019) and fragmentation of the mitochondrial network in hypothalamic neurons (Schneeberger *et al*, 2013; Santoro *et al*, 2017). Hyperleptinemia may itself stem from excessive mitochondria fission; eliminating the mitochondrial fusion mediator MFN2 from adipocytes is sufficient to produce hyperleptinemia and obesity (Mancini *et al*, 2019). Thus, therapies that preserve normal mitochondrial morphology and function in the brain and white adipose tissue should improve leptin responsiveness via complementary mechanisms. Reversing mitochondrial fragmentation is a particularly appealing strategy because it may also improve insulin resistance and hepatic steatosis (Jheng *et al*, 2012; Sebastián *et al*, 2012; Galloway *et al*, 2014; Wang *et al*, 2015; Filippi *et al*, 2017). In sum, our poorly stocked armamentarium and the expanding scope of the obesity epidemic provide a strong impetus to develop and test innovative therapeutic approaches such as limiting mitochondrial fission.

In mice with high-fat diet (HFD)-induced obesity, mitochondrial fragmentation occurs downstream of increased C16:0 ceramide production (Chavez & Summers, 2012; Choi & Snider, 2015; Hammerschmidt *et al*, 2019; Summers *et al*, 2019). The Western diet is high in saturated fat which leads to increased circulating levels of palmitate, supplying both backbone and fatty acid chain for C16:0 ceramide synthesis. Reducing ceramide production by deleting serine palmitoyl transferase (Li *et al*, 2011), ceramide synthase 6 (Turpin *et al*, 2014; Hammerschmidt *et al*, 2019), or dihydroceramide desaturase 1 (Holland *et al*, 2007; Chaurasia *et al*, 2019) protects mice from the negative metabolic consequences of consuming a HFD in part by preventing fragmentation of the mitochondrial network. Unfortunately, drugs that safely and selectively target these enzymes are not yet available. Drugs that block mitochondrial fission downstream of ceramide production could produce many of the same benefits as limiting ceramide production. Agents reported to target the mitochondrial fission factor DRP1 or to up-regulate the fusion factor MFN2 have been evaluated in obesity models (Jheng *et al*, 2012; Ayanga *et al*, 2016; Filippi *et al*, 2017; Chen *et al*, 2018). However, these compounds have produced either no or very limited benefits in mice with diet-induced obesity and failed to trigger significant weight loss. Here, we provide a potential explanation: these molecules either do not block ceramide-induced mitochondrial fission or are effective only after prolonged incubation periods. We therefore evaluated an alternative, indirect approach to targeting mitochondrial dynamics, disrupting endolysosomal trafficking. Lysosomal proteins play an essential role in mitochondrial fission by recruiting the dynamin-related GTPase DRP1 (Abuarab *et al*,

2017; Wong *et al*, 2018; Peng *et al*, 2020), while endocytic recycling proteins promote fission through a mechanism that is incompletely defined (Farmer *et al*, 2017). Consistent with its ability to interfere with both endosomal recycling and lysosomal trafficking (Kim *et al*, 2016; Finicle *et al*, 2018), the orally bioavailable, water-soluble synthetic sphingolipid SH-BC-893 blocked palmitate- and ceramide-induced mitochondrial fission both in vitro and in vivo. Importantly, unlike previously tested molecules, SH-BC-893 produced weight loss and metabolic improvements consistent with published genetic studies that identified excessive mitochondrial fission as a key driver of diet-induced obesity.

## Results

### SH-BC-893 protects from palmitate- and ceramide-induced mitochondrial fragmentation under conditions where other compounds are ineffective

In mice consuming a HFD, increased circulating palmitate is converted to C16:0 ceramide, triggering fragmentation of the mitochondrial network and the negative metabolic consequences associated with obesity (Jheng *et al*, 2012; Sebastián *et al*, 2012; Schneeberger *et al*, 2013; Smith *et al*, 2013; Wang *et al*, 2015; Filippi *et al*, 2017; Hammerschmidt *et al*, 2019). Murine embryonic fibroblasts (MEFs) possess a highly tubular mitochondrial network and are thus commonly used to monitor mitochondrial fission. As expected, palmitate supplementation produced dramatic mitochondrial fragmentation in MEFs (Figs 1A–D and EV1) (Hammerschmidt *et al*, 2019). Similar results were obtained in MEFs exposed to C16:0 ceramide (Fig 1E–H). C16:0 ceramide triggers the recruitment of the DRP1 GTPase that mediates fission to mitochondria (Hammerschmidt *et al*, 2019). The small molecule mdivi-1 (Cassidy-Stone *et al*, 2008) has been employed as an inhibitor of DRP1 in obesity models. Mdivi-1 reduces ROS production in palmitate- or ceramide-treated C2C12 myotubes (Smith *et al*, 2013), moderately improves insulin resistance without affecting glucose clearance in *ob/ob* mice (Jheng *et al*, 2012), restores insulin-mediated suppression of hepatic glucose production in HFD-fed rats (Filippi *et al*, 2017), and slows the progression of diabetic nephropathy in *db/db* mice (Ayanga *et al*, 2016). However, compelling evidence suggests that the limited benefits of mdivi-1 in obesity models actually stem from mitochondrial complex I inhibition rather than direct inhibition of DRP1 (Bordt *et al*, 2017); the type 2 diabetes mellitus therapy metformin is also a complex I inhibitor. Consistent with this hypothesis, mdivi-1 purchased from two different suppliers failed to prevent C16:0 ceramide-induced mitochondrial fragmentation even at elevated doses, after a prolonged pre-incubation, or when combined with the small molecule mitochondrial fusion promoter M1 as in (Buck *et al*, 2016) (Fig EV2A–D). The inability of mdivi-1 to preserve mitochondrial morphology in ceramide-treated cells indicates that this compound cannot be used to gauge the potential therapeutic value of opposing mitochondrial fission in patients with obesity.

Other compounds that modulate mitochondrial dynamics were similarly evaluated. The dihydroorotate dehydrogenase inhibitor leflunomide triggers transcriptional up-regulation of the mitochondrial fusion factors MFN1 and MFN2 (Miret-Casals *et al*, 2018) and has been tested in murine obesity models. Consistent with its

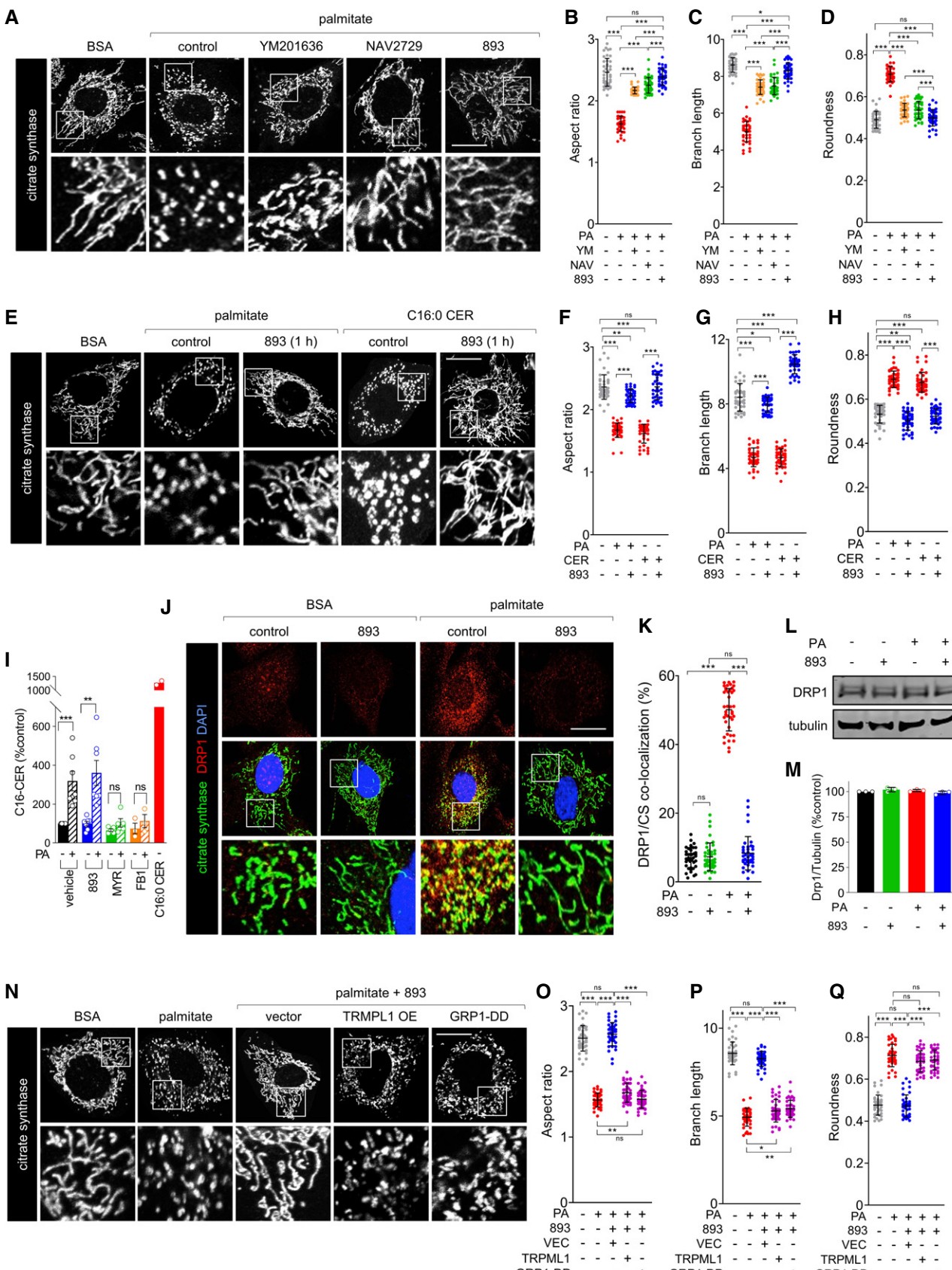

**Figure 1.**

**Figure 1. SH-BC-893 protects from ceramide-induced mitochondrial network fragmentation by disrupting intracellular trafficking.**

A  Citrate synthase staining in MEFs treated for 3 h with BSA or palmitate (250 μM) after a 3-h pre-treatment with DMSO (control), YM201636 (YM, 800 nM), NAV2729 (NAV, 12.5 μM), or SH-BC-893 (893, 5 μM).

B–D  ImageJ was used to calculate aspect ratio (B), branch length (C), and roundness (D) of mitochondria in the cells in (A) as described in Fig EV1. PA, palmitate.

E  MEFs were treated with BSA, palmitate (250 μM), or C16:0 CER (100 μM) for 3 h after a 1-h pre-treatment with vehicle or SH-BC-893 (5 μM) and mitochondria stained as in (A).

F–H  Aspect ratio (F), branch length (G), and roundness (H) of mitochondria in the cells in (E).

I  Mean C16:0 ceramide levels in MEFs pre-treated for 3 h with vehicle (n = 7), SH-BC-893 (5 μM, n = 7), myriocin (myr, 10 μM, n = 5), or fumonisin B1 (FB1, 30 μM, n = 3) then treated with BSA or palmitate (250 μM) for 3 h. Cells treated with C16:0 CER (100 μM, n = 2) for 3 h shown as a positive control. Error bars, SEM.

J  MEFs treated as in (A) but evaluated for DRP1 (red) and citrate synthase (green) co-localization (yellow) using confocal immunofluorescence microscopy. Nuclei are labeled with DAPI (blue).

K  Mander's overlap coefficient for DRP1 and citrate synthase (CS) for the cells in (J) calculated on a per cell basis.

L, M  Representative DRP1 Western blot (L) or quantification of DRP1 levels (M) using cells treated as in (J); n = 3. Mean ± SD.

N  MEFs expressing vector, TRPML1, or GRP1-DD were treated with BSA or palmitate (250 μM) for 3 h and stained as in (A).

O–Q  Aspect ratio (O), branch length (P), and roundness (Q) of mitochondria in the cells in (N).

Data information: In B-D, F-H, K, and O-Q, 40 cells from 2 biological replicates were evaluated and mean ± SD shown. Using a one-way ANOVA with Tukey's correction (C, D, H, M, P and Q), Brown-Forsythe and Welch ANOVA tests with Dunnett's correction for multiple comparisons (B, F, G, K and O), or unpaired, two-tailed $t$-tests (I), ***$P \leq 0.001$; **$P \leq 0.01$; *$P \leq 0.05$; ns, not significant, $P > 0.05$ (key comparisons shown). Scale bars, 20 μm.

mechanism of action, leflunomide blocked C16:0 ceramide-induced mitochondrial fragmentation after a 24-h, but not a 1-h, pre-incubation (Fig EV2E–H). The need for prolonged exposure may explain the inability of leflunomide to correct HFD-induced obesity (Chen et al, 2018). Similar to leflunomide, the cell-permeant peptide inhibitor of DRP1, P110 (Qi et al, 2013), was effective only after a prolonged incubation (Fig EV2I–L). Although not previously evaluated for their effects on mitochondrial morphology, the compounds celastrol and withaferin A have been reported to reduce ER stress, sensitize to leptin, and protect from HFD-induced obesity (Liu et al, 2015; Lee et al, 2016), outcomes that would be consistent with reduced mitochondrial fission. However, at the concentrations used in these earlier studies, neither celastrol nor withaferin A preserved mitochondrial networks, and celastrol triggered severe mitochondrial fragmentation even in the absence of palmitate (Fig EV2M–P). Together, these results expose an unmet need for small molecules that robustly and acutely block ceramide-induced mitochondrial fission.

Targeting endolysosomal trafficking could offer an alternative strategy to disrupt ceramide-induced mitochondrial fragmentation. The recruitment of DRP1 to mitochondria depends on contact with lysosomes (Wong et al, 2018). The lysosomal cation channels TRPML1 and TRPM2 promote the fission process (Abuarab et al, 2017; Li et al, 2017; Peng et al, 2020). TRPML1 channel activity is regulated by PI(3,5)P$_2$, a lipid produced from PI(3)P by the PIKfyve kinase (Dong et al, 2010). Consistent with these relationships, the PIKfyve inhibitor YM201636 significantly reduced palmitate-induced mitochondrial fragmentation (Fig 1A–D). The recycling endosome protein EHD1 also promotes mitochondrial fission; deleting EHD1 produces a highly tubulated mitochondrial network (Farmer et al, 2017). EHD1 is positively regulated by the ARF6 GTPase (Caplan et al, 2002) suggesting that, like EHD1 knockdown, ARF6 inhibition might block mitochondrial fragmentation. Indeed, the ARF6 inhibitor NAV2729 preserved a tubular mitochondrial network in palmitate-treated MEFs (Fig 1A–D). We have previously shown that the synthetic sphingolipid SH-BC-893 disrupts endolysosomal trafficking by both inhibiting the PIKfyve pathway (Kim et al, 2016) and inactivating ARF6 (Finicle et al, 2018), parallel actions that might robustly oppose mitochondrial fission in the face of rising ceramide levels. In fact, SH-BC-893 provided rapid and complete protection from palmitate-induced mitochondrial fission (Fig 1A–

H). Consistent with the expectation that its effects lie downstream of ceramide synthesis, SH-BC-893 did not inhibit ceramide synthesis and was similarly effective in cells exposed to C16:0 ceramide (Fig 1E–I). In keeping with its proposed mechanism of action, SH-BC-893 blocked palmitate-induced recruitment of DRP1 to mitochondria without affecting DRP1 protein levels (Fig 1J–M). Moreover, PIKfyve pathway and ARF6 inhibition are necessary for these effects as restoring endolysosomal trafficking by over-expressing TRPML1 (Kim et al, 2016) or introducing a phosphomimetic mutant of the ARF6 GEF, GRP1-DD (Finicle et al, 2018), each undermined the protection offered by SH-BC-893 (Fig 1N–Q). In contrast to ceramide-induced mitochondrial fission, SH-BC-893 did not prevent fragmentation induced by uncoupling with FCCP (Fig EV2Q–T). In summary, disrupting endolysosomal trafficking with the synthetic sphingolipid SH-BC-893 rapidly and robustly blocked palmitate- and ceramide-induced mitochondrial fragmentation.

### SH-BC-893 preserves mitochondrial function in palmitate- or ceramide-treated cells

Mitochondrial fragmentation leads to functional deficits that contribute to obesity. Fragmented mitochondria generate more ROS and induce ER stress (Yu et al, 2006; Schrepfer & Scorrano, 2016); ER stress has been linked to leptin resistance and obesity (Zhang et al, 2008; Ozcan et al, 2009; Schneeberger et al, 2013; Liu et al, 2015). As expected, palmitate treatment profoundly reduced mitochondrial membrane potential (Fig 2A and B). Co-incubation with SH-BC-893 blocked this effect. Ceramide-induced ROS generation was also blocked by SH-BC-893 (Fig 2C and D). Consistent with these improvements in mitochondrial function, SH-BC-893 prevented the palmitate-induced ER stress response as measured by Chop, Xbp1s, Atf6, and Grp78 mRNA levels (Fig 2E). These experiments demonstrate that, at least in vitro, SH-BC-893 protects from the changes in mitochondrial form and function that have been linked to HFD-induced obesity.

### SH-BC-893 preserves mitochondrial morphology and function in the tissues of mice with HFD-induced obesity

To determine whether SH-BC-893 also protects from ceramide-induced mitochondrial fragmentation in vivo, male C57BL/6J mice

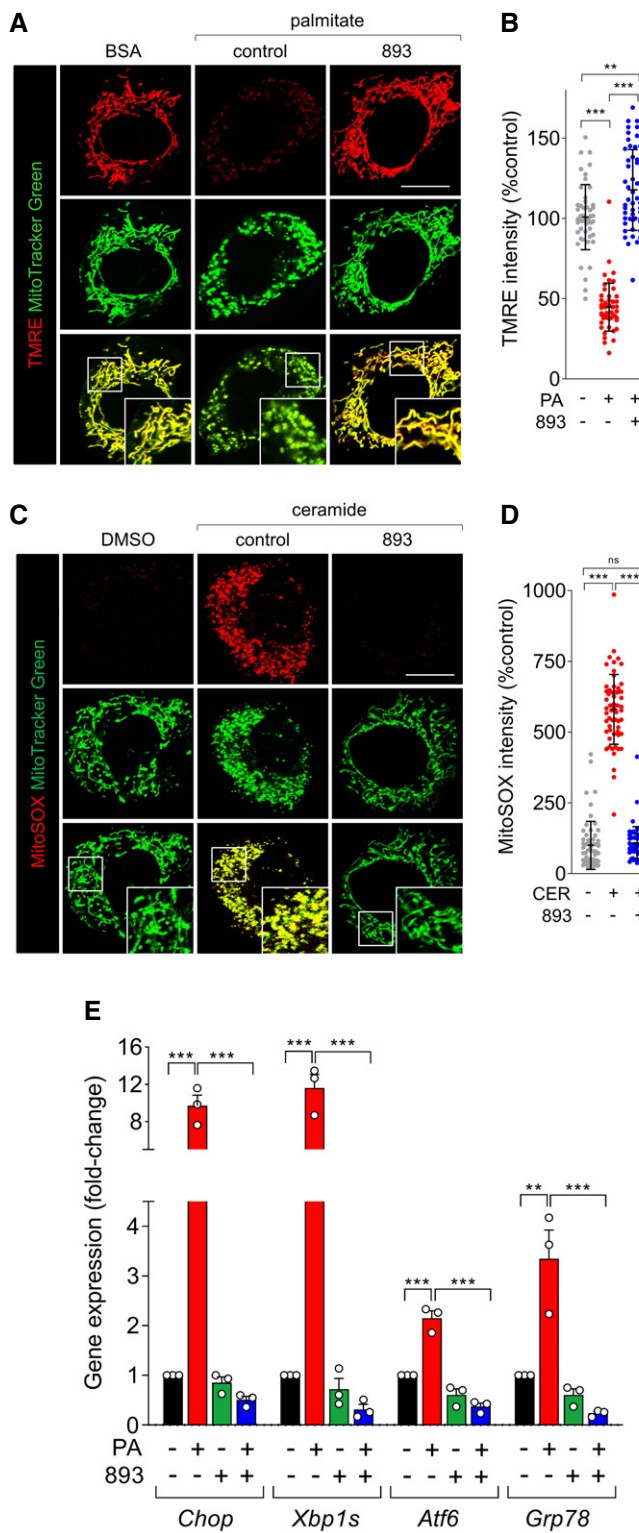

**Figure 2. SH-BC-893 protects from ceramide-induced mitochondrial dysfunction.**

A TMRE (100 nM) and MitoTracker Green (200 nM) staining in MEFs treated with BSA or palmitate (250 μM) for 12 h after a 3-h pre-treatment with vehicle or SH-BC-893 (893, 5 μM).

B TMRE intensity in (A) on a per cell basis.

C MitoSOX (5 μM) and MitoTracker Green (200 nM) staining in MEFs treated with vehicle (DMSO) or C2-ceramide (50 μM) for 12 h after a 3-h pre-treatment with vehicle or SH-BC-893 (893, 5 μM).

D MitoSOX intensity for (C) on a per cell basis.

E *Chop, Xbp1s, Atf6,* and *Grp78* mRNA levels measured in MEFs treated for 16 h with BSA or palmitate (250 μM) after a 3-h pre-treatment with vehicle or SH-BC-893 (893, 5 μM); $n = 3$. Mean ± SEM.

Data information: In B and D, 50-60 cells from 2 biological replicates were evaluated and mean ± SD shown. Using Brown–Forsythe and Welch ANOVA tests with Dunnett's correction (B, D) or a one-way ANOVA with Tukey's correction for multiple comparisons (E), ***$P \leq 0.001$; **$P \leq 0.01$; ns, not significant, $P > 0.05$ (key comparisons shown). Scale bars, 20 μm.

groups (Fig 3A). When assessing mitochondrial networks, light microscopy has two significant advantages over electron microscopy: (i) the 3D architecture of mitochondria is directly visualized rather than inferred and (ii) unbiased measurements of mitochondrial shape can be made in a large number of cells using image analysis software (Fig EV3A). Furthermore, evaluating intact, viable tissue by light microscopy avoids artifacts introduced by fixation or lengthy cell isolation procedures. Freshly resected liver was refractory to staining with MitoTracker Green, but NAD(P)H autofluorescence allowed visualization of mitochondrial networks (Fig 3B). Notably, NAD(P)H autofluorescence and MitoTracker Green produced identical staining patterns in MEFs (Fig EV3B), and mitochondrial morphology measured using NAD(P)H autofluorescence was consistent with images of hepatic mitochondria in livers from transgenic mice expressing mitochondrially directed fluorescent proteins or tags (Shitara *et al*, 2010; Jacobi *et al*, 2015; Barrasso *et al*, 2018; Bayraktar *et al*, 2019). Mitochondrial morphology varies over the circadian cycle (Jacobi *et al*, 2015); mitochondria become maximally fragmented several hours into the dark cycle (ZT12-ZT24) when mice begin feeding. When evaluated between ZT13 and ZT17, mitochondria in the livers of HFD-fed mice were larger and more spherical than in mice consuming the SD (Fig 3B–D). Based on the pharmacokinetics of SH-BC-893 ($t_{max} = 4$ h, $t_{1/2} = 10.6$ h), animals were gavaged with vehicle (water) or 120 mg/kg SH-BC-893 at ZT8.5, 3.5 h before the onset of the dark period. This dose of SH-BC-893 is effective and well tolerated in prostate cancer models (Kim *et al*, 2016). A single treatment with SH-BC-893 normalized the morphology of hepatic mitochondria in mice consuming the HFD, increasing mitochondrial tubularity (aspect ratio) and reducing roundness to match SD controls (Fig 3B–D). SH-BC-893 did not significantly alter hepatic mitochondrial morphology in lean mice consuming the SD. These results demonstrate that SH-BC-893 can acutely restore normal mitochondrial morphology in vivo in HFD-fed mice with obesity.

Mitochondrial fragmentation in the hypothalamus contributes to leptin resistance and obesity (Schneeberger *et al*, 2013; Santoro *et al*, 2017). Due to the increased time required to remove and section the brain, mitochondrial morphology was examined in this tissue using immunofluorescence microscopy rather than by NAD (P)H autofluorescence. Consistent with the abnormal mitochondrial

were fed a 45% kcal from fat rodent diet (HFD) for 24 weeks starting at 6 weeks of age and liver mitochondrial morphology evaluated by confocal microscopy. An aged-matched cohort of mice fed a standard chow diet (SD, 10% kcal from fat) was produced in parallel. As expected, body weights were significantly different in these

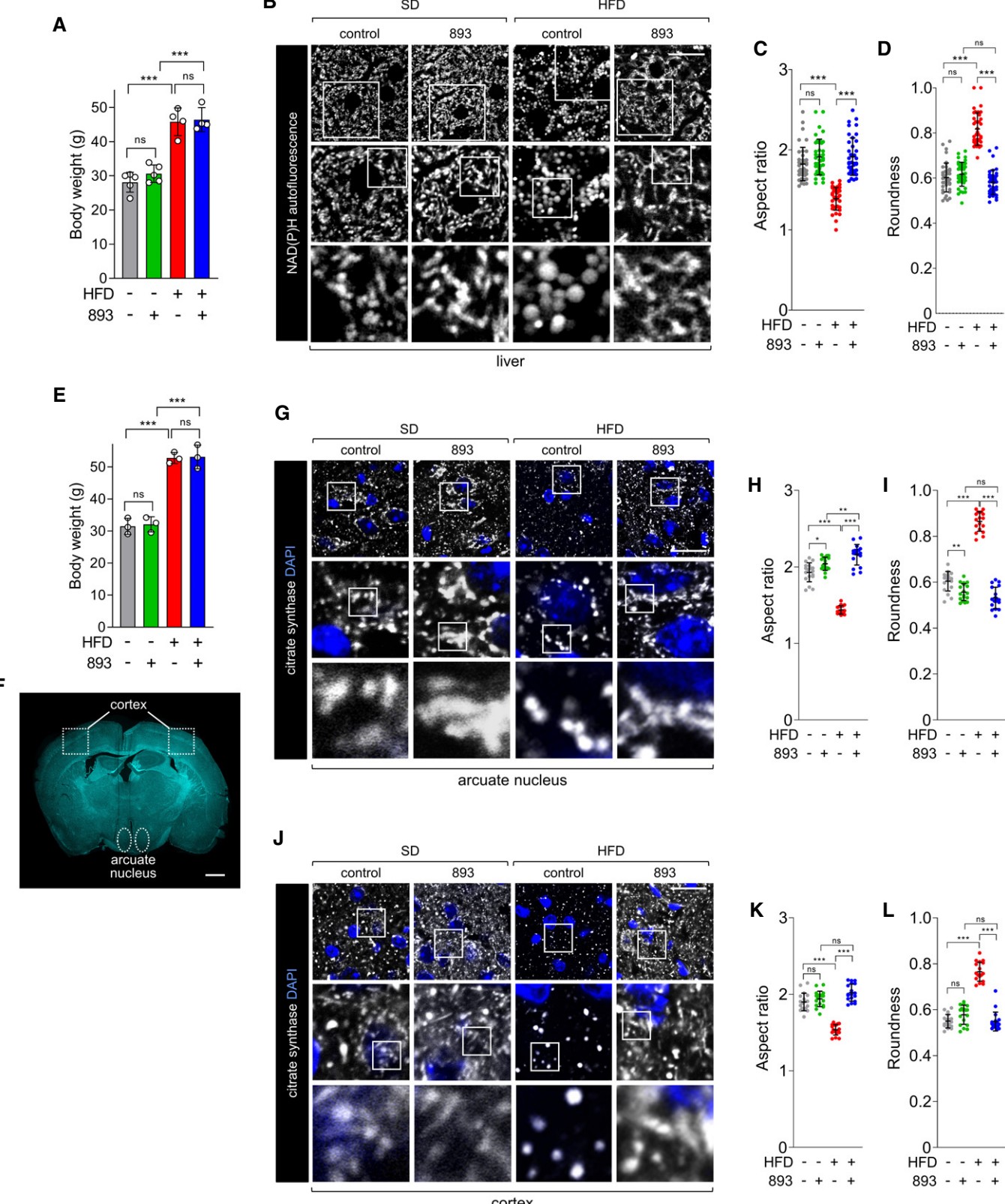

**Figure 3.**

**Figure 3.  SH-BC-893 acutely restores a tubular mitochondrial network in the liver and brain of HFD-fed mice.**

A  Body weights of mice that had consumed a SD for 22 weeks or a HFD for 26 weeks; *n* = 4–5. Mean ± SD. Mice are grouped to reflect their randomization to vehicle or SH-BC-893 groups, weight measured prior to treatment.

B  NAD(P)H autofluorescence evaluated by confocal microscopy in freshly resected livers from mice in (A) 4–8 h after treatment with vehicle (*n* = 5) or 120 mg/kg SH-BC-893 (*n* = 4) by gavage at ZT8.5. Mice were sacrificed and imaged in pairs (SD and HFD).

C, D  Aspect ratio (C) and roundness (D) of mitochondria in the livers shown in (B) calculated per field as described in Fig EV3; 40 fields examined from each treatment group.

E  Body weights of mice that had consumed a chow or HFD diet for 22 weeks; *n* = 3. Mean ± SD. Mice are grouped to reflect their randomization to vehicle or SH-BC-893 groups, weight measured prior to treatment.

F  Coronal brain section showing the areas where mitochondria were imaged in G-L.

G  Citrate synthase staining in the arcuate nucleus of mice in (E) 4 h after treatment with vehicle or 120 mg/kg SH-BC-893 by gavage at ZT8.5; *n* = 3.

H, I  Aspect ratio (H) or roundness (I) of the mitochondria in (G); 18 fields examined from each treatment group.

J–L  As in G-I, but imaging was performed in the areas of the cortex outlined in (F). 18 fields examined from each treatment group.

Data information: In A, C, D, E, H, I, K and L mean ± SD shown; 8-12 (C, D) or 6 (H, I, K and L) fields evaluated per mouse. Using a one-way ANOVA with Tukey's correction for multiple comparisons, \*\*\*$P \leq 0.001$; \*\*$P \leq 0.01$; \*$P \leq 0.05$; ns, not significant, $P > 0.05$ (key comparisons shown). Scale bars, 20 µm (B, G, and J) or 1 mm (F).

morphology in the livers of HFD-fed mice (Fig 3B–D), mitochondria in the arcuate nucleus of the hypothalamus of mice with HFD-induced obesity were less tubular and rounder (Fig 3E–I). Similar changes were observed in the cerebral cortex of the same mice (Fig 3J–L). In sum, within 4 h of oral administration, SH-BC-893 acutely reversed HFD-induced changes in mitochondrial morphology in both liver and brain.

Mitochondrial dysfunction in white adipose tissue (WAT) also contributes to obesity (Kusminski & Scherer, 2012). Deleting the gene encoding the mitochondrial fusion protein MFN2 selectively in adipocytes produces hyperleptinemia, reduces circulating adiponectin levels, and makes mice as overweight on a chow diet as wild-type animals become on a HFD (Mancini *et al*, 2019). In mice maintained on either diet, mitochondrial morphology was difficult to resolve in eWAT and quantification was not possible, but neither HFD diet nor SH-BC-893 produced gross alterations in mitochondrial shape (Fig 4A and B). However, NAD(P)H autofluorescence was significantly reduced in the eWAT of HFD-fed mice relative to SD controls, and this deficit was rapidly and robustly corrected by SH-BC-893 (Fig 4A–C). As expected (Kusminski & Scherer, 2012; Kusminski *et al*, 2016), the mitochondrial membrane potential was severely reduced in the white adipocytes of mice with HFD-induced obesity relative to lean controls (Fig 4D–F). Remarkably, this defect was also corrected 4 h after treatment with SH-BC-893. In contrast, although the mitochondrial mass per cell increased with HFD feeding and adipocyte hypertrophy as expected, it was not affected by SH-BC-893 (Fig 4G). Also in keeping with published studies (Kusminski & Scherer, 2012; Kusminski *et al*, 2016), mitochondrial ROS production in eWAT was elevated by the HFD (Fig 4H–J). This defect was also corrected by SH-BC-893. Consistent with these improvements in mitochondrial function, repeated dosing with SH-BC-893 reduced ER stress in eWAT (Fig 4K). Taken together, these results indicate that SH-BC-893 rapidly and robustly restores mitochondrial form and/or function in the tissues of mice with HFD-induced obesity.

## SH-BC-893 corrects obesity-associated alterations in circulating leptin and adiponectin

Published studies link unbalanced mitochondrial fission in WAT to increased leptin production and decreased adiponectin secretion (Mancini *et al*, 2019). ER stress may contribute to these changes (Kusminski & Scherer, 2012). The improvements in mitochondrial function in the WAT of HFD-fed mice treated with SH-BC-893 (Fig 4) suggested that adipokine secretion might also be normalized. As expected, plasma adiponectin levels were low and leptin levels were elevated in mice fed a HFD for 8 weeks (Fig 5A and B). Surprisingly, SH-BC-893 increased circulating adiponectin and reduced plasma leptin within 4 h of administration. It has been proposed that the ratio of adiponectin to leptin is a more accurate predictor of metabolic health and better measure of WAT dysfunction than either adipokine in isolation. In human patients, an adiponectin:leptin ratio $\geq 1.0$ is considered healthy, $0.5 – 1.0$ considered moderate risk, while values $< 0.5$ are associated with a high risk of lethal comorbidities such as cardiovascular disease and cancer (Gong *et al*, 2015; Frühbeck *et al*, 2018; Frühbeck *et al*, 2019). A single dose of SH-BC-893 raised the mean adiponectin:leptin ratio in HFD-fed mice from 0.15 to 0.59 (Fig 5C). This improvement persisted with repeated dosing with reductions in leptin driving the increased ratio (Fig 5D–F). Hyperleptinemia is sufficient to produce leptin resistance (Zhao *et al*, 2019). Chronic elevations in circulating leptin trigger negative feedback loops in the hypothalamus, including up-regulation of the negative regulator of STAT signaling, SOCS3, and down-regulation of the leptin receptor, LEPR. A decline in *Socs3* mRNA and increase in *Lepr* mRNA in the hypothalamus of SH-BC-893 treated mice (Fig 5G and H) suggest that the observed reductions in plasma leptin (Fig 5B and E) were sufficient to interrupt this negative feedback and restore central leptin sensitivity. Mice with progressively severe obesity remained responsive to SH-BC-893. Similar fold-changes in adiponectin, leptin, and the adiponectin:leptin ratio were observed 4 h after SH-BC-893 administration in mice maintained on the HFD for 4–16 weeks (Fig 5I–L). Together, correction of plasma leptin and adiponectin levels and improvement in markers of central leptin resistance suggest that SH-BC-893 could be effective as an interventional agent for diet-induced obesity.

## SH-BC-893 reduces food intake and triggers weight loss

Excessive mitochondrial fission drives leptin resistance indirectly by promoting hyperleptinemia and directly by reducing the sensitivity of POMC neurons to leptin signaling (Schneeberger *et al*, 2013; Santoro *et al*, 2017; Zhao *et al*, 2019). By normalizing mitochondrial

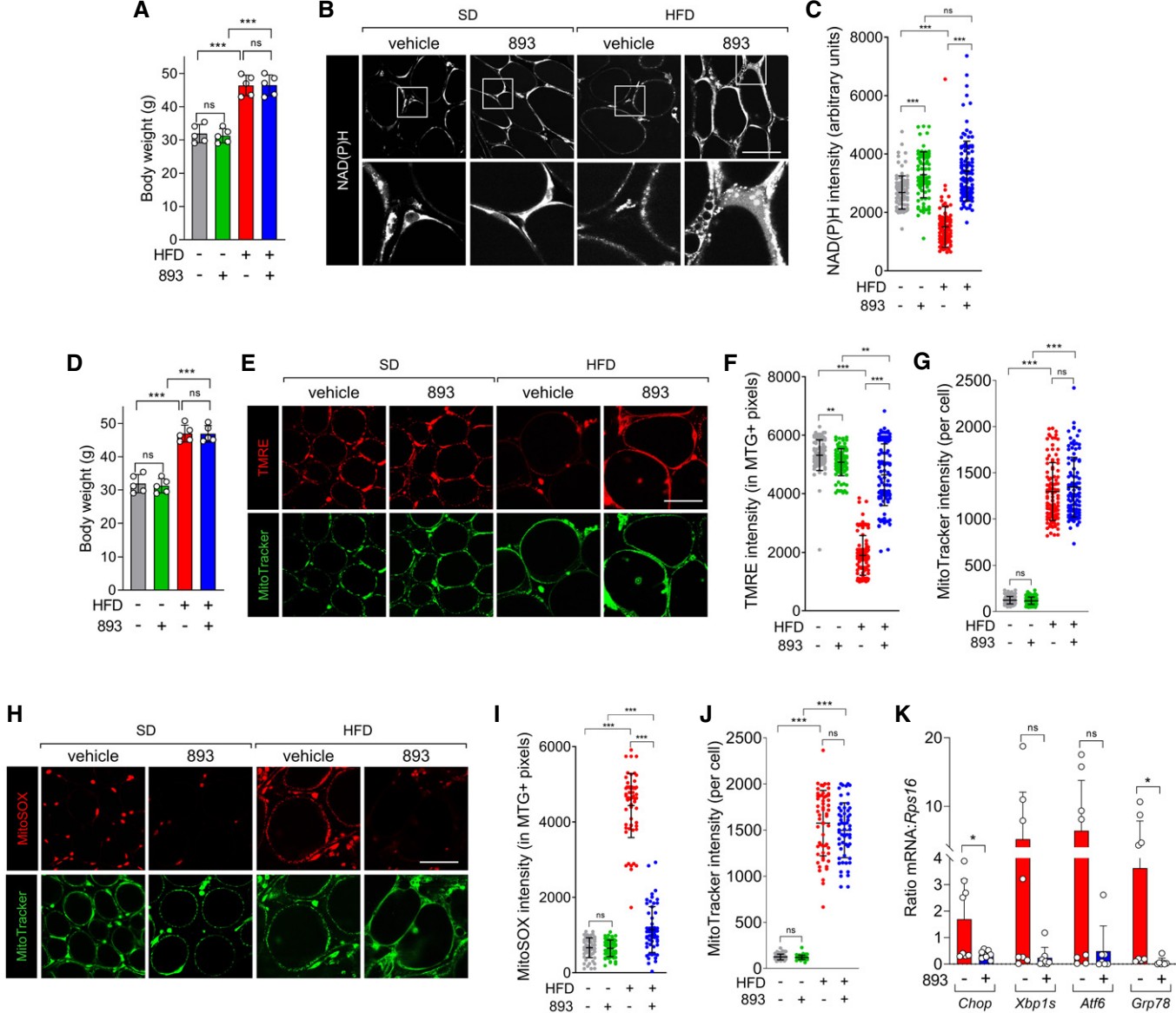

**Figure 4. SH-BC-893 preserves mitochondrial function in the WAT of mice with HFD-induced obesity.**

A   Body weight of mice fed the SD or HFD for 16 weeks; *n* = 5. Mice are grouped to reflect their randomization to vehicle or 120 mg/kg SH-BC-893 groups, weight measured prior to treatment.

B   NAD(P)H autofluorescence was evaluated in epididymal fat (eWAT) freshly resected from the mice in (A) 4 h after treatment with vehicle or 120 mg/kg SH-BC-893 by gavage at ZT8.5; *n* = 5.

C   NAD(P)H intensity from (B) calculated on a per cell basis; 16–25 cells from each of the 5 mice per group were quantified (87–109 total cells per treatment group).

D   Body weight of mice fed the SD or HFD for 16 weeks; *n* = 5. Mice are grouped to reflect their randomization to vehicle or 120 mg/kg SH-BC-893 groups, weight measured prior to treatment.

E   TMRE (100 nM) and MitoTracker Green staining (200 nM) in eWAT resected from mice in (D) treated as in (B, C); *n* = 5.

F, G   TMRE (F) or MitoTracker Green (G) intensity for (E) calculated on a per cell basis; 15–24 cells from each of the 5 mice in each group were quantified (99–104 total cells per treatment group).

H   MitoSOX (5 μM) and MitoTracker Green staining in eWAT resected from the mice in (D) treated as in (B and E); *n* = 5.

I, J   MitoSOX (I) or MitoTracker Green (J) intensity calculated as in (F and G); 10–13 cells from each of the 5 mice in each group were quantified (57–58 total cells per treatment group).

K   *Chop*, *Xbp1s*, *Atf6*, and *Grp78* mRNA measured in eWAT resected from mice maintained on a HFD for 13 weeks and gavaged with vehicle or 120 mg/kg SH-BC-893 on Monday, Wednesday, and Friday for the final 3 weeks; *n* = 7–9.

Data information: In A, C, D, F, G, and I-K mean ± SD shown. Using a one-way ANOVA and Tukey's correction for multiple comparisons (A, D), Brown–Forsythe and Welch ANOVA tests with Dunnett's correction for multiple comparisons (C, F, G, I, and J), or unpaired *t*-tests with Welch's correction (K), ***$P \leq 0.001$; **$P \leq 0.01$; *$P \leq 0.05$; ns, not significant, $P > 0.05$. Scale bars, 20 μm.

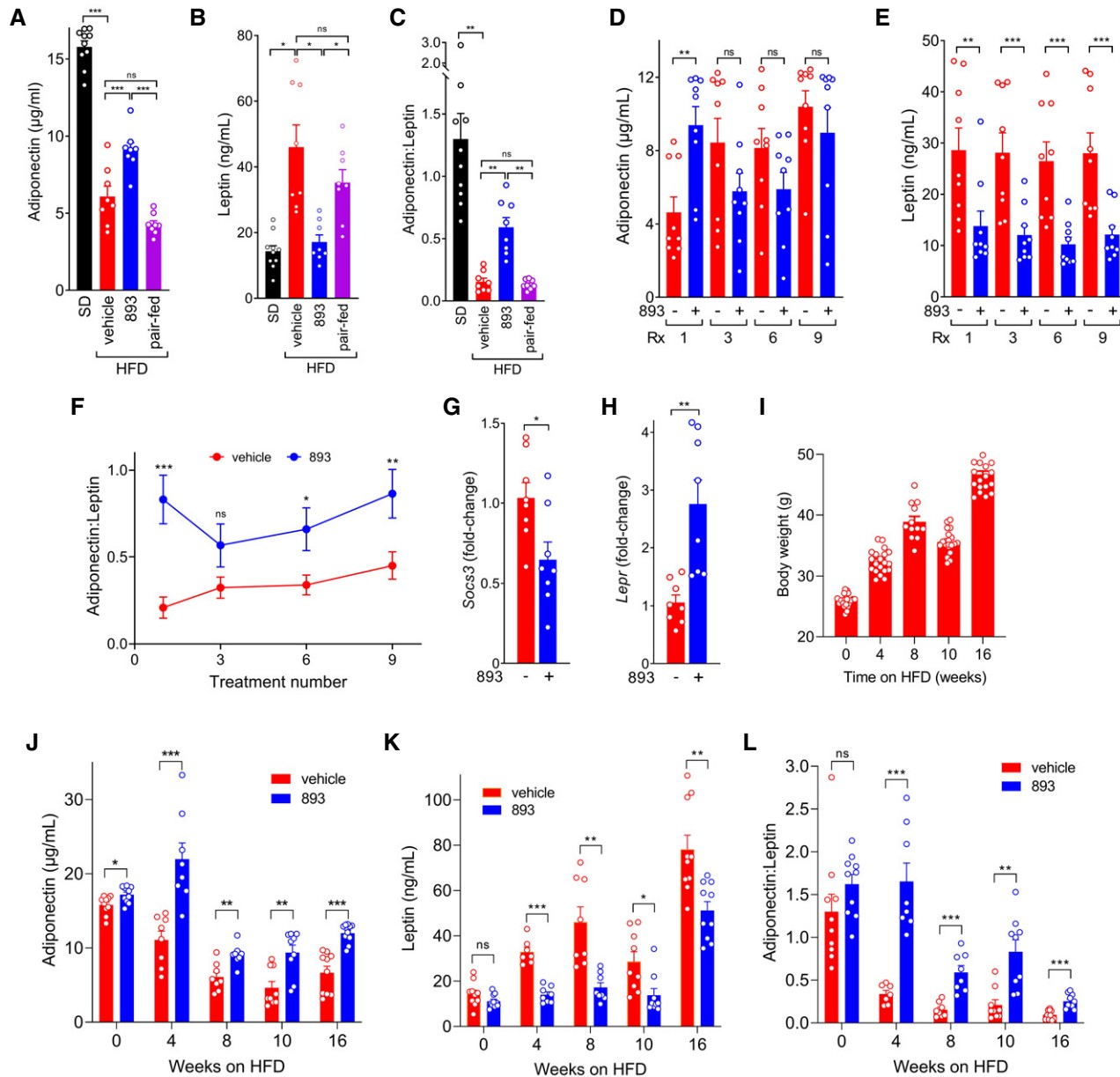

**Figure 5. SH-BC-893 normalizes adipokine levels in mice consuming a HFD.**

A, B    Plasma adiponectin (A) or leptin (B) in mice maintained on a SD or HFD for 8–9 weeks and then gavaged with vehicle or 120 mg/kg SH-BC-893 4 h prior to blood collection. Pair-fed mice were provided with the amount of food consumed by SH-BC-893-treated mice in 4 h but left untreated; $n$ = 8–10.

C       The adiponectin:leptin ratio was calculated for the mice in A, B.

D, E    As in (A, B) but in mice maintained on the HFD for 10 weeks and then gavaged Mondays, Wednesdays, and Fridays for 3 weeks with vehicle or 120 mg/kg SH-BC-893; $n$ = 9. Rx, treatment number of a total of nine doses.

F       The adiponectin:leptin ratio was calculated for the mice in (D, E); $n$ = 9.

G, H    Mice were treated as in D-F, sacrificed 4 h after the ninth dose, and the levels of *Socs3* (G) or *Lepr* (H) mRNA measured in the hypothalamus; $n$ = 8.

I       Body weights of mice fed the HFD for the indicated interval. Two cohorts were utilized: one measured at 4 and 8 wk and the other at 10 and 16 weeks; $n$ = 16–20.

J, K    Plasma adiponectin and leptin were measured in the mice in (I) 4 h after a single dose of 120 mg/kg SH-BC-893; $n$ = 7–10.

L       The adiponectin:leptin ratio was calculated for the mice in J,K; $n$ = 7–10.

Data information: Mean ± SEM shown. Using a one-way ANOVA with Tukey's correction (A), Brown–Forsythe and Welch ANOVA tests with Dunnett's correction for multiple comparisons (B, C), or unpaired two-tailed $t$-tests (D-H and J-L, Welch's correction applied in H), ***$P$ ≤ 0.001; **$P$ ≤ 0.01; *$P$ ≤ 0.05; ns, not significant, $P$ > 0.05.

form and/or function in the hypothalamus and WAT (Figs 3E–I and 4), SH-BC-893 should restore leptin sensitivity through parallel mechanisms, reducing food intake. Consistent with this model and

SH-BC-893's pharmacokinetics, a single dose reduced food intake in mice with HFD-induced obesity for about 18 h (Fig 6A and B). Mice were similarly sensitive to a second dose, and animals did not

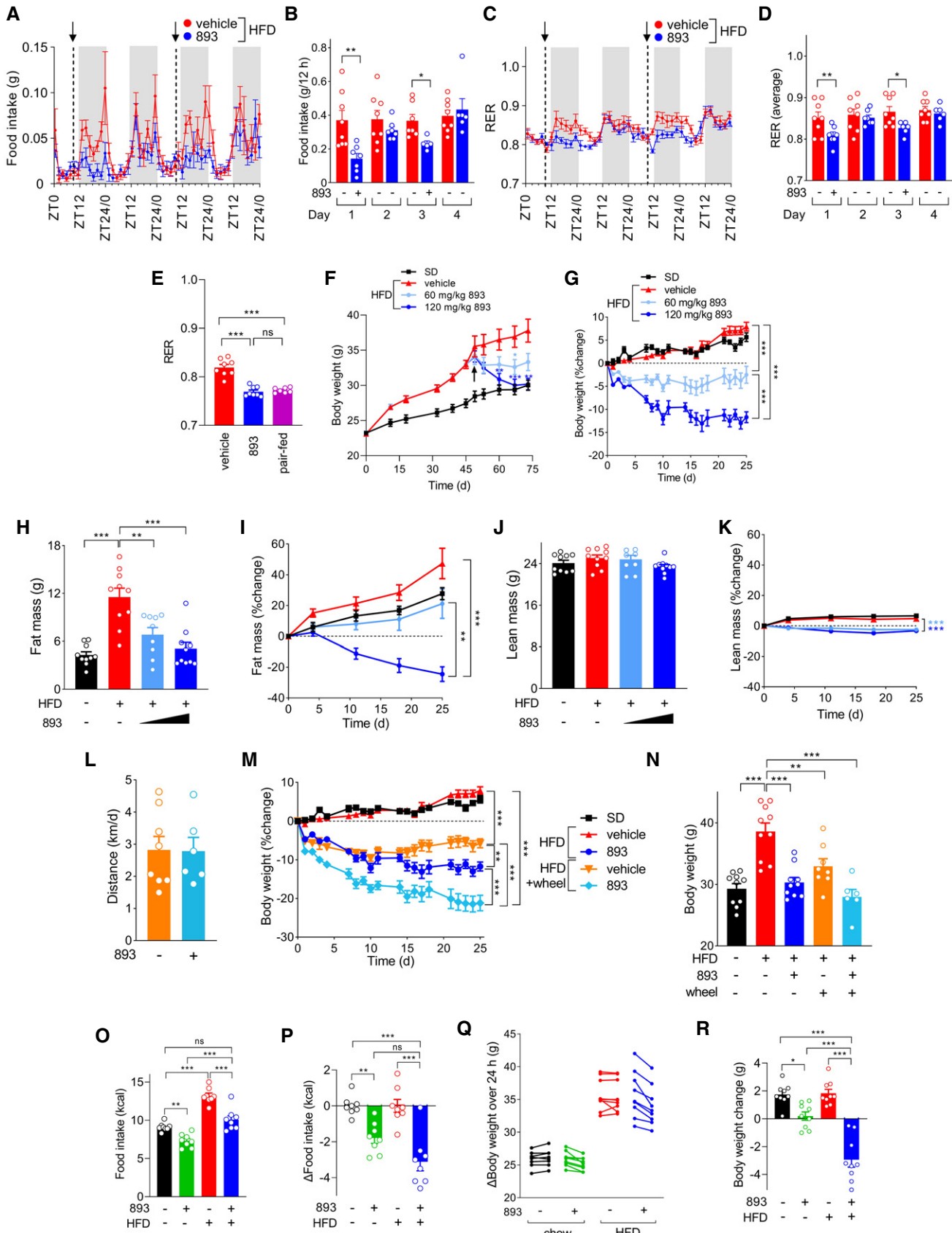

**Figure 6.**

**Figure 6. SH-BC-893 reduces food intake and triggers weight loss.**

A, B   Food intake during indirect calorimetry studies in mice fed the HFD for 10–12 weeks and then gavaged with vehicle or 120 mg/kg SH-BC-893 at ZT8.5 on days 1 (first exposure) and 3; *n* = 5–8. The means of 4 measurements over 108 min (A) or over the dark cycle (ZT12-ZT24) (B) shown ± SEM. In (A), arrows indicate dosing. Gray shading indicates dark period.

C, D   As in (A, B) but measuring the respiratory exchange ratio (RER); *n* = 6–8.

E      Average RER from ZT12-ZT24 in mice fed the HFD for 22 weeks and then treated once at ZT8.5 with vehicle or 120 mg/kg SH-BC-893; *n* = 7–8. Pair-fed mice were fed the amount of food eaten by SH-BC-893-treated animals.

F, G   Absolute (F) or relative (G) body weight of mice fed the standard diet and gavaged with vehicle (SD, *n* = 10) or a HFD and gavaged with vehicle (*n* = 10), 60 mg/kg (*n* = 9), or 120 mg/kg (*n* = 10) SH-BC-893 on Mondays, Wednesdays, and Fridays beginning on day 47 (arrow in F).

H–K    Fat (H, I) or lean (J, K) mass in the mice in (F) measured after the 4 weeks of treatment (H, J) or weekly (I, K); *n* = 9–10.

L      Average daily distance run by HFD-fed mice treated with vehicle (*n* = 8) or SH-BC-893 (120 mg/kg, *n* = 6) on Mondays, Wednesdays, and Fridays over 4 weeks when housed with a running wheel.

M, N   Body weight of mice fed the SD or HFD housed with or without wheels and treated with vehicle or 120 mg/kg SH-BC-893 as indicated over time (M) or after 4 weeks of treatment (N); *n* = 6–10.

O, P   Mice were fed chow or the HFD for 9–10 weeks then treated with vehicle or 120 mg/kg SH-BC-893 Mondays, Wednesdays, and Fridays for 3 weeks; *n* = 9. Mean absolute (O) or relative (P) caloric intake per mouse in the 24 h after treatment is plotted for each of 8 different treatments; mice were euthanized 4 h after the ninth treatment.

Q, R   Body weight change in the mice in (O, P) over the 24 h after the first treatment (Q) or after the 3 week course of treatment (R); *n* = 9.

Data information: In all panels, mean ± SEM shown. Using unpaired *t*-tests (B, D, and L), a Brown–Forsythe and Welch ANOVA test with Dunnett's correction for multiple comparisons (E, F day 73, and I), or a one-way ANOVA with Tukey's correction (other days in F, G, H, J, K, M, N, O, P, and R), *$P \leq 0.05$; **$P \leq 0.01$; ***$P \leq 0.001$; ns, not significant, $P > 0.05$. In G, I, K, and M, statistical analysis was performed using only the data from the last day of the experiment.

compensate by increasing food intake on the day following treatment. Importantly, the reduction in circulating leptin likely drives rather than follows from reduced food intake as neither leptin nor adiponectin levels improved in pair-fed mice that were provided with only the reduced amount of food eaten by SH-BC-893-treated animals (Fig 5A and B). Parallel reductions in RER (Fig 6C and D) likely reflect increased catabolism of stored fat in response to reduced food intake rather than a primary change in metabolic programming as pair feeding fully recapitulated the effects of SH-BC-893 on RER (Fig 6E). Energy expenditure calculated using the Weir formula was not significantly affected by SH-BC-893 (Fig EV4A). A trend toward reduced activity as measured by XY beam breaks (Fig EV4B) may relate to decreased food seeking behavior. These results indicate that SH-BC-893-induced reductions in food intake are sufficient to trigger the catabolism of stored fat.

SH-BC-893 was next evaluated as an interventional agent in group-housed mice. After 47 days on the HFD, animals were randomly assigned to receive vehicle or SH-BC-893 at 60 mg/kg or 120 mg/kg by gavage on Mondays, Wednesdays, and Fridays for 4 weeks. While the vehicle-treated HFD group continued to gain weight as expected, mice treated with 60 mg/kg or 120 mg/kg SH-BC-893 exhibited dose-dependent weight loss despite continued consumption of the HFD (Fig 6F and G). Despite continued treatment with the high dose of SH-BC-893, weight loss plateaued once body weight matched that of SD controls (Fig 6F). The dose-dependent weight loss in SH-BC-893-treated mice was primarily due to a decline in fat mass with little change in lean mass indicating that overall body composition was improved (Fig 6H–K). Mice treated with 60 mg/kg SH-BC-893 gained fat mass at a similar rate to mice fed a standard diet (Fig 6I) indicating that this dose was sufficient to prevent adiposity resulting from HFD feeding. As in our prior report where mice were dosed 5–7 days a week for 11 weeks (Kim *et al*, 2016), SH-BC-893 was well tolerated, and the behavior of SH-BC-893-treated mice was overtly normal throughout the study. Thus, SH-BC-893 restores normal adiposity and body weight in previously obese mice despite the continuous feeding of a HFD.

Medical therapies for obesity are generally coupled with lifestyle interventions. To determine whether the beneficial effects of SH-BC-893 on body weight and adiposity were additive with voluntary exercise, a subset of HFD-fed mice were provided with running wheels. As rodent running activity declines under stress (Garland *et al*, 2011), monitoring the duration and distance of voluntary wheel running also provides a holistic measure of overall mouse health. HFD-fed mice receiving vehicle ran an average daily distance of 2.8 ± 0.7 km over the course of the experiment, a value that was not significantly different from the SH-BC-893-treated group (2.8 ± 1.2 km, Fig 6L). The average time spent on running wheels each day was also equivalent in vehicle- and SH-BC-893-treated groups; exercise activity was generally well-matched between the groups on a given day suggesting that day-to-day differences in activity were likely related to uncontrolled variations in the environment (Fig EV4C–E). HFD-fed mice receiving vehicle and housed with a running wheel exhibited similar reductions in adiposity and body weight as SH-BC-893-treated mice maintained in standard caging (Figs 6 M and N, and EV4F and G). Mice both provided with a running wheel and treated with SH-BC-893 exhibited even greater reductions in body weight and adiposity than produced by either treatment alone without loss of lean mass (Figs 6 M and N, and EV4F–I). In sum, SH-BC-893 reduced body weight and adiposity additively with, and to a similar extent as, voluntary exercise.

Although SH-BC-893 did not significantly reduce circulating leptin levels in SD-fed animals (Fig 5K), it did increase the tubularity and reduce the roundness of hypothalamic mitochondria in SD-fed controls (Fig 3G–I). Thus, SH-BC-893 might reduce food intake even in the absence of high-fat feeding. In keeping with this proposal, eliminating DRP1 from POMC neurons increases responsiveness to leptin and other satiety signals even in chow-fed mice (Santoro *et al*, 2017). Consistent with this report, SH-BC-893 also reduced caloric intake in chow-fed mice (Figs 6O and P, and EV4J and K). Despite eating less, SH-BC-893-treated chow-fed mice maintained their body weight on the same SH-BC-893 treatment schedule that produced significant weight loss in mice with HFD-induced

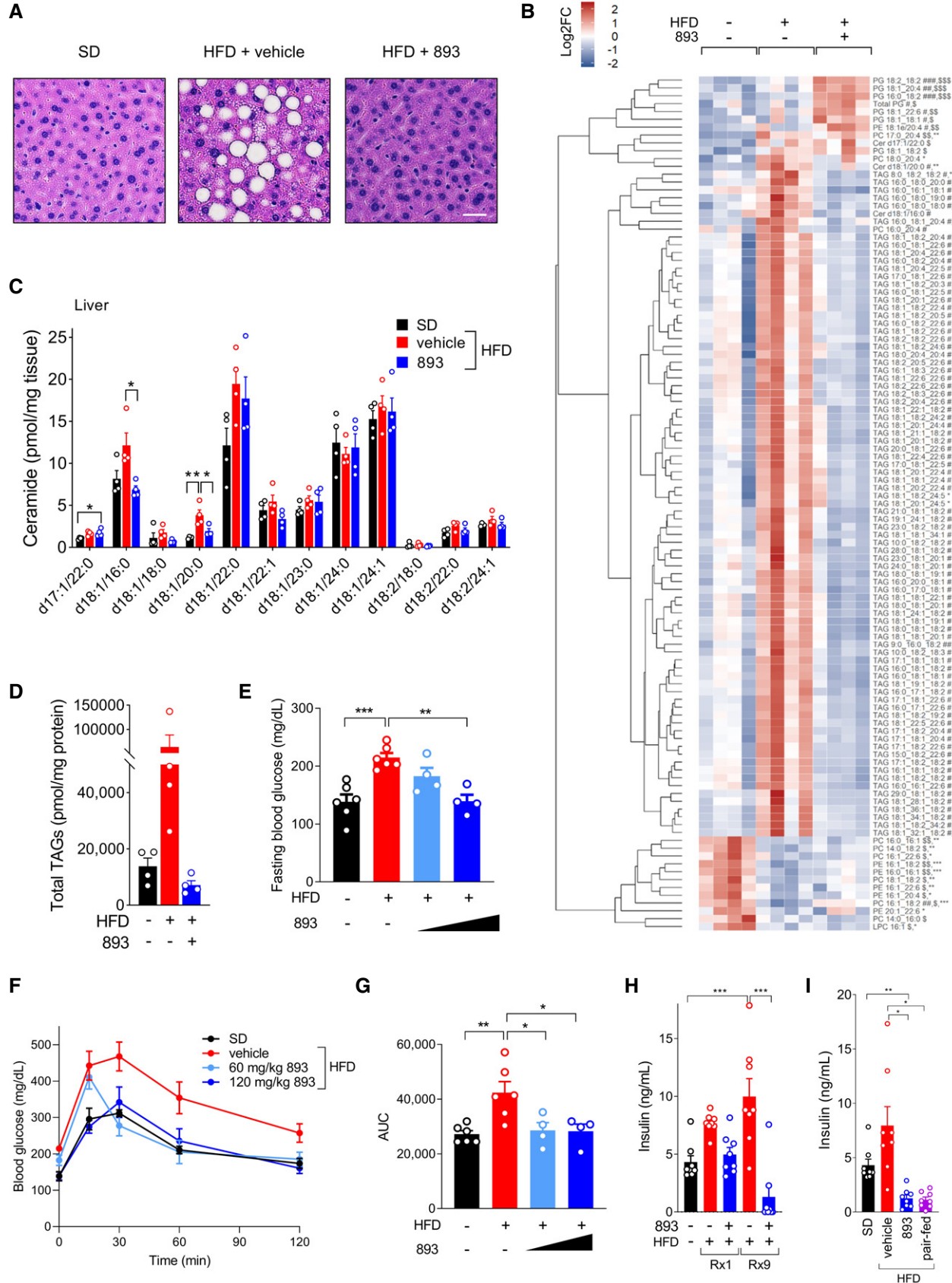

**Figure 7.**

**Figure 7. SH-BC-893 corrects metabolic dysfunction associated with HFD-induced obesity.**

A Representative H&E-stained liver sections from the mice shown in Fig 6F after 4 weeks of treatment with vehicle or 120 mg/kg SH-BC-893. Scale bar, 10 μm.

B Unbiased lipidomics performed on the livers of 4 randomly selected mice from Fig 6F. Lipids that were significantly different in SD, HFD + vehicle, or HFD + 120 mg/kg SH-BC-893 shown. PG, phosphatidylglycerol; PE, phosphatidylethanolamine; PC, phosphatidylcholine; Cer, ceramide; TAG, triacylglycerol; and LPC, lysophosphatidylcholine.

C Ceramide levels in livers of the same mice as in (B); $n = 4$.

D Total triacylglycerol (TAG) levels in livers of the same mice as in (B); $n = 4$.

E Fasting blood glucose from mice randomly selected from the cohorts in Fig 6F after 25 days of treatment; $n = 4$–6.

F, G Blood glucose levels (F) or area under the curve (AUC, (G)) during an oral glucose tolerance test performed in the same mice as in (E).

H Insulin levels in plasma collected 4 h after the indicated treatment with vehicle or SH-BC-893 from the mice in Fig 5D and E or in 9-week-old SD-fed mice treated with vehicle (same mice as in Fig 5J and K, 0 weeks on HFD); $n = 8$.

I Insulin levels in the plasma collected 4 h after treatment with vehicle or SH-BC-893 from the same mice as in Fig 5A and B; $n = 8$.

Data information: Mean ± SEM shown. Using a one-way ANOVA with Tukey's correction (B-E, G and H) or a Brown–Forsythe and Welch ANOVA test with Dunnett's correction for multiple comparisons (I), $*P \leq 0.05$; $**P \leq 0.01$; $***P \leq 0.001$. In B, comparisons are made between SD and HFD (*), HFD + vehicle and HFD + 120 mg/kg SH-BC-893 (#), or SD and HFD + 120 mg/kg SH-BC-893 ($).

obesity (Fig 6Q and R). Thus, SH-BC-893 appears to decrease food intake in lean mice to the level required for body weight homeostasis; C57BL6/J male mice fed chow ad libitum eat in excess of their caloric needs leading to progressive increases in adipose tissue (Fig 6I). This hypothesis is consistent with the flattening of the body weight loss curve once HFD-fed animals treated with 120 mg/kg SH-BC-893 reached normal weight (Fig 6F). In sum, SH-BC-893 does not cause progressive weight loss in non-obese controls or formerly obese mice, but instead appears to improve coupling between ad libitum caloric intake and the energy demands associated with maintaining an optimal body weight.

### SH-BC-893 corrects the metabolic disruptions associated with HFD consumption

Because metabolic health is not always correlated with body mass, whether SH-BC-893 could correct the negative metabolic sequelae associated with HFD consumption was assessed. Chronic HFD feeding leads to the ectopic accumulation of lipids in non-adipose tissues (lipotoxicity). Excessive hepatic lipid accumulation can eventually lead to liver fibrosis and inflammation, increasing the risk of hepatocellular carcinoma (Haas *et al*, 2016). Weight loss is a highly effective treatment for non-alcoholic fatty liver disease, but SH-BC-893 may also improve fatty liver disease by reducing mitochondrial fission in hepatocytes (Fig 3B–D, Galloway *et al*, 2014) and increasing adiponectin levels (Fig 5A and J, Xu *et al*, 2003). Indeed, both histology and unbiased lipidomics confirmed that SH-BC-893 resolved lipotoxicity in mice consuming the HFD, producing trends toward lower triacylglyceride levels in the liver and ceramide levels in the liver and muscle (Figs 7A–D and EV5A). Obesity is also linked to type 2 diabetes mellitus. SH-BC-893 could improve glucose handling in mice consuming a HFD through parallel mechanisms: directly via mitochondrial fusion in liver, muscle, and pancreatic beta cells and indirectly by increasing circulating adiponectin and/ or reducing body weight (Jheng *et al*, 2012; Kusminski & Scherer, 2012; Sebastián *et al*, 2012; Stiles & Shirihai, 2012; Turer & Scherer, 2012; Xia *et al*, 2014; Wang *et al*, 2015). Indeed, SH-BC-893 normalized both fasting glucose and the response to an oral glucose tolerance test (OGTT) in HFD-fed mice (Fig 7E–G). SH-BC-893 also reduced plasma insulin levels, both after a single dose and after 3 weeks of treatment (Fig 7H). However, insulin levels appear to decline secondary to decreased food intake as paired

feeding recapitulated the effects of SH-BC-893 (Fig 7I). In sum, the results presented here establish SH-BC-893 as an effective intervention for both HFD-induced obesity and its negative metabolic sequelae.

## Discussion

Here, we show that the small molecule SH-BC-893 acutely reversed HFD-induced changes in mitochondrial morphology and/or function in liver, brain, and WAT (Figs 3 and 4), producing weight loss (Fig 6F and G) and normalizing metabolism (Fig 7) despite continued consumption of a HFD. It is likely that these benefits stem primarily from a reduction in caloric intake (Figs 6A, B, O and P and EV4J and K). SH-BC-893 likely decreases food intake through parallel mechanisms that lie downstream of reduced mitochondrial fission and ER stress, increasing hypothalamic sensitivity to multiple anorexigenic stimuli (Zhang *et al*, 2008; Schneeberger *et al*, 2013; Santoro *et al*, 2017) and correcting hyperleptinemia (Zhao *et al*, 2019) (Fig EV5B). Excessive mitochondrial fission reduces sensitivity to insulin in multiple tissues (Jheng *et al*, 2012; Wang *et al*, 2015; Filippi *et al*, 2017). As insulin signaling in the brain influences feeding behavior (Kullmann *et al*, 2016), SH-BC-893 may also enhance satiety by overcoming central insulin and leptin resistance stemming from the ER stress and inflammation associated with overnutrition (Zhang *et al*, 2008). Indirect effects on the hypothalamus are also likely in play. Improving mitochondrial function in WAT resolves hyperleptinemia in HFD-fed mice (Fig 5B, E and K), curtailing the negative feedback that limits leptin signaling in the hypothalamus (Fig 5G and H) (Zhao *et al*, 2019). Genetically engineered mouse models where *Mfn2* is deleted in POMC neurons and/or adipose tissue could help to parse the extent to which central and peripheral effects of the compound are necessary and/or sufficient to reduce food intake. Stereotactic injection of SH-BC-893 into the brain might also help dissect the compound's central and peripheral actions. Together, the activation of these anorexigenic pathways reduces caloric intake and makes SH-BC-893 an effective interventional agent in mice with HFD-induced obesity.

Although studies in genetically engineered mouse models have firmly established excessive mitochondrial fission as a root cause of HFD-induced obesity (Fig EV5B), small molecules reported to block

mitochondrial fission failed to produce significant weight loss in HFD-fed mice (Jheng *et al*, 2012; Filippi *et al*, 2017; Chen *et al*, 2018). The suboptimal activity of these agents in ceramide-treated cells in vitro (Fig EV2) and the robust weight loss in SH-BC-893-treated animals (Fig 6F and G) suggest that mitochondrially targeted agents have failed in pre-clinical models due to their lack of efficacy rather than a flawed therapeutic premise. Indeed, the published studies employing mdivi-1 and leflunomide in mice with obesity did not include rigorous pharmacodynamic studies of mitochondrial morphology in multiple tissues as was performed here with SH-BC-893 (Figs 3 and 4). Both leflunomide and the peptide inhibitor of DRP1, P110, were effective in vitro after prolonged pre-incubations (Fig EV2E–L). However, the failure of leflunomide in mice with HFD-induced obesity (Chen *et al*, 2018) suggests that a delayed mode of action may be incompatible with drug pharmacokinetics. Moreover, as a peptide therapeutic, P110 would require parenteral administration; an oral drug like SH-BC-893 would be much more suitable as an interventional agent in patients with obesity. In sum, this work establishes that SH-BC-893 is superior to existing agents used to modulate mitochondrial morphology and that targeting ceramide-induced mitochondrial fission with a small molecule can safely and effectively combat HFD-induced obesity.

SH-BC-893's mechanism of action, disrupting endocytic trafficking events (Fig EV5B), has not previously been tested as a means to therapeutically modulate mitochondrial fission. Lysosomes are key regulators of mitochondrial morphology; about 15% of LAMP1-positive vesicles are in contact with mitochondria at a given time in HeLa cells, marking sites of DRP1 recruitment and eventual fission (Wong *et al*, 2018). By disrupting PIKfyve-dependent lysosomal trafficking (Kim *et al*, 2016), SH-BC-893 likely interferes with TRPML1-mediated $Ca^{2+}$ and/or $Zn^{2+}$ transfer from lysosomes to mitochondria, a required step for fission (Abuarab *et al*, 2017; Li *et al*, 2017; Peng *et al*, 2020). How ARF6 inhibition suppresses mitochondrial fission is uncertain (Fig 1B–E), but inactivation of the ARF6 effector EHD1 is a likely explanation (Farmer *et al*, 2017). While individual ARF6 and PIKfyve inhibitors also oppose ceramide-induced mitochondrial fission, SH-BC-893's dual actions on both pathways make it more effective (Fig 1A–D). SH-BC-893 most likely disrupts these intracellular trafficking steps by activating PP2A (Kim *et al*, 2016; Finicle *et al*, 2018) (Fig EV5B). Although several structurally distinct PP2A agonists have been described (Ruvolo, 2016), other PP2A agonists may not recapitulate the anti-obesogenic effects of SH-BC-893 due to distinct modes of PP2A activation and/or their significant off-target actions (Lee & Choi, 2016). Many PP2A agonists inactivate AKT (Gutierrez *et al*, 2014; Suman *et al*, 2016; Tohmé *et al*, 2019; Tsuji *et al*, 2021) which makes them ill-suited for use as obesity therapies as they would cause insulin resistance. SH-BC-893 does not reduce AKT phosphorylation (Kubiniok *et al*, 2019), most likely because it activates PP2A through a different mechanism than other agonists. Given that PP2A activation will have multiple downstream effects apart from alterations in endolysosomal trafficking, additional studies will be required to determine the extent to which mitochondria-independent actions contribute to SH-BC-893's anti-obesogenic effects. In sum, SH-BC-893 has a unique mode of action that merits more complete dissection in follow-up studies.

Increasing adiponectin levels is an important therapeutic goal given that it improves insulin sensitivity, reduces inflammation, and improves metabolic function in obesity models (Xu *et al*, 2003; Kim *et al*, 2007; Gastaldelli *et al*, 2010; Phillips & Kung, 2010). Humans with obesity that have high levels of adiponectin are at lower risk of myocardial infarction, liver and kidney disease, and endometrial cancer (Turer & Scherer, 2012; Frühbeck *et al*, 2018). Adiponectin receptor agonists have been reported (Okada-Iwabu *et al*, 2013), and thiazolidinediones (TZDs) both increase adiponectin and decrease leptin levels (Fasshauer & Paschke, 2003). While SH-BC-893 consistently reduced leptin levels, its ability to boost plasma adiponectin may be limited to the initial dose (Fig 5D). However, alternative dosing strategies may be helpful given that a single dose reproducibly elevates circulating adiponectin in animals with different degrees of obesity (Fig 5J). While increasing adiponectin is not likely to contribute to the reduction in calorie intake (Figs 6A, B, O and P and EV4J and K; Xu *et al*, 2003; Asterholm & Scherer, 2010; Okada-Iwabu *et al*, 2013)), it may contribute to the improved metabolic health of SH-BC-893-treated animals (Fig 7). For example, although SH-BC-893 does not limit ceramide production from palmitate (Fig 1I), adiponectin receptors possess ceramidase activity (Holland *et al*, 2011), and thus, increased adiponectin may contribute to reductions in tissue ceramides (Figs 7B and C and EV5A). Increases in circulating adiponectin have been linked to the ability of TZDs to limit the progression of NAFLD to NASH (Gastaldelli *et al*, 2010; Warshauer *et al*, 2015; Cusi *et al*, 2016; Li *et al*, 2019). However, determining whether SH-BC-893 would limit the progression of fatty liver disease will require chronic HFD feeding in combination with fructose and/or evaluation of NAFLD-prone mouse strains. In sum, the ability of SH-BC-893 to elevate circulating adiponectin levels may have therapeutic value independent of the weight loss that occurs secondary to leptin sensitization.

SH-BC-893 was initially developed as an anti-cancer agent (Kim *et al*, 2016). It is interesting to speculate that its antineoplastic activity may stem in part from its ability to oppose mitochondrial fission. Mitochondrial networks are fragmented in most cancer cells, and reversing this phenotype can block tumor growth (Serasinghe *et al*, 2015; Senft & Ronai, 2016; Lima *et al*, 2018; Nagdas *et al*, 2019). SH-BC-893 also reverses mitochondrial fragmentation driven by aberrant KRAS activation (Appendix Fig S1A–H). Notably, SH-BC-893 was also more effective than leflunomide at producing a tubulated mitochondrial network in cancer cells with activated KRAS, a model system where leflunomide has been shown to have antineoplastic activity (Yu *et al*, 2019). Furthermore, SH-BC-893 is able to oppose mitochondrial fission in response to signals other than ceramide given that blocking ceramide synthesis was ineffective in KRAS-mutant cells (Appendix Fig S1I–L). Incidentally, mdivi-1 and M1 were also unable to oppose KRAS-dependent mitochondrial fission. The observation that SH-BC-893 not only slows autochthonous prostate tumor growth, but also maintains tumors in a more differentiated state (Kim *et al*, 2016) might be linked to its effects on mitochondria. Mitochondrial fission promotes stemness and maintains tumor-initiating cells (Katajisto *et al*, 2015; Xie *et al*, 2015). Thus, SH-BC-893 may target cancer stem cells by maintaining a tubulated mitochondrial network. As tumor cell migration requires trafficking of mitochondria to the leading edge and mitochondrial transport depends on fission (Senft & Ronai, 2016), SH-BC-893 might additionally subvert cancer metastasis by blocking mitochondrial fission. Obesity itself promotes cancer through multiple mechanisms (Park *et al*, 2014; Ringel *et al*, 2020), and thus, SH-BC-893's

ability to reduce caloric intake (Fig 6A and B) and adiposity (Fig 6H) may synergize with its direct, cell-autonomous effects on cancer cells (Chen *et al*, 2016; Kim *et al*, 2016; Finicle *et al*, 2018; Kubiniok *et al*, 2019). The growth of many tumors is spurred by insulin (Gallagher & LeRoith, 2020), and resolution of hyperinsulinemia by SH-BC-893 (Fig 7H and I) could also limit tumor growth. Finally, adiponectin levels are inversely and leptin levels directly correlated with prostate cancer grade in some cohorts (Saglam *et al*, 2003; Goktas *et al*, 2005; Karnati *et al*, 2017) suggesting that effects of SH-BC-893 on these adipokines (Fig 5) may contribute to its ability to suppress prostate cancer growth (Kim *et al*, 2016). Given these connections between mitochondrial morphology, adipokines, obesity, and cancer, it will be important to assess the effects of SH-BC-893 on mitochondria in tumors and its effectiveness in tumor models that are sensitive to a HFD and/or leptin and insulin levels.

While SH-BC-893 is a highly effective intervention in C57BL6/J male mice with HFD-induced obesity (Figs 6 and 7), it remains unclear whether it will be similarly effective in humans with diet-induced obesity. However, unbalanced mitochondrial fission has been linked to increased adiposity and metabolic dysfunction in humans as well as mice. Patients homozygous for a missense mutation in *MFN2* have fragmented, spherical adipocyte mitochondria and a dramatic upper body adipose tissue over-growth syndrome (Rocha *et al*, 2017). Large, round mitochondria have also been observed in pancreatic β cells of patients with type 2 diabetes (Anello *et al*, 2005). It will be interesting to determine whether SH-BC-893 prevents mitochondrial fragmentation in additional metabolic tissues such as pancreatic β cells and muscle cells where excessive fission during obesity has also been linked to metabolic deficits (Jheng *et al*, 2012; Sebastián *et al*, 2012). Whether SH-BC-893 will be effective in females, in other models of diet-induced obesity (e.g., HFD + fructose), and whether it will suppress hedonic food intake are important questions given that the answers will have therapeutic implications. Intriguingly, human trials with natural sphingolipids are consistent with the results reported here in SH-BC-893-treated mice. Dietary supplementation with the structurally related fungal sphingolipid phytosphingosine improves glucose metabolism in humans as well as HFD-fed mice (Snel *et al*, 2010; Murakami *et al*, 2013). It is encouraging that SH-BC-893 further improves the response to exercise (Figs 6 M and N, and EV4F and G) as clinical trials for obesity therapies often combine new agents with diet and lifestyle interventions (Adler, 2021). In sum, the available information suggests that SH-BC-893, and potentially natural sphingolipids, could be effective therapies for obesity in human patients.

A medical therapy for obesity must meet higher safety standards than a cancer therapy. Indeed, many obesity programs have been discontinued due toxicity (Sam *et al*, 2011). While it may be surprising that an agent that affects mitochondrial morphology in a wide array of tissues is well tolerated, it is important to consider that the effects of drugs wax and wane as their tissue levels rise and fall while the effects of genetic manipulations are constitutive. Although a more in-depth characterization of SH-BC-893's toxicity profile will be an important next step, unchanged voluntary wheel running (Figs 6L and EV4C–E), observations that adiponectin reduces depressive behaviors in mice (Yau *et al*, 2014), and reports linking depression to mitochondrial fragmentation in multiple cell types (Fan *et al*, 2019; Gebara *et al*, 2020) suggest that SH-BC-893 may not only avoid the negative psychiatric side effects of previous

appetite suppressants (Sam *et al*, 2011), but actually prove beneficial. Importantly, even more frequent administration of SH-BC-893 for a longer interval does not adversely affect rapidly dividing cells in the bone marrow and gut or cause organ dysfunction (Kim *et al*, 2016). In conclusion, the current study offers an important proof of concept showing that targeting mitochondrial fission is safe, feasible, and effective in a diet-induced obesity model, results that might translate to other human diseases where the pathophysiology is driven by fragmentation of the mitochondrial network (Archer, 2013; Schrepfer & Scorrano, 2016; Li *et al*, 2019; Chan, 2020).

# Materials and Methods

## General animal procedures

All animal experiments were performed in accordance with the Institutional Animal Care and Use Committee of University of California, Irvine. Six- to eight-week-old, male C57BL/6J mice (stock no 000664) were purchased from the Jackson Laboratory and acclimated for 7 days prior to use in experiments. Mice were housed under a 12-h:12-h light–dark cycle at 20–22°C in groups of 4–5. Cages contained 1/8" corncob bedding (7092A, Envigo, Huntingdon, UK) enriched with ~6 g of cotton fiber nestlets (Ancare Corp., Bellmore, NY). Mice were fed the vivarium stock diet (chow) which contained 16% kcal from fat (2020x, Envigo). For HFD studies, mice were fed a 45% kcal from fat diet (HFD; D12451, Research Diets Inc., New Brunswick, NJ) or a matched, standard diet (SD) containing 10% kcal from fat diet (SD; D12450B, Research Diets Inc) for 4–26 weeks as indicated in the figure legends. Access to food and water was ad libitum unless otherwise specified. Polypropylene feeding tubes (20 g × 38 mm; Instech Laboratories Inc., Plymouth, PA) or sterilized metal stainless steel gavage needles (20G, CAD7910, Sigma-Aldrich) were utilized for gavage and dipped into a 1 g/ml sucrose solution immediately prior to treatment to induce salivation. On rare occasions (3 times out of > 500 gavage doses), animals required euthanasia due to gavage errors. Inadvertent pharyngeal administration of gavage solution during removal of the feeding tube occurred in 4 cases but did not require euthanasia. Data from these animals were censored.

## Intervention studies

In intervention studies conducted without running wheels, mice were communally housed (3-5 to a cage) and randomly assigned to experimental groups. Treated mice received either vehicle (water) or SH-BC-893 at 60 mg/kg or 120 mg/kg by oral gavage as a single dose or on Mondays, Wednesdays, and Fridays and maintained on the HFD throughout the study. In Figs 6F–N and EV4F–I, the SD group was treated with vehicle on the M/W/F schedule starting on day 47. In this intervention study, group size was initially $n = 10$; one animal from the 60 mg/kg group that was euthanized due to gavage error was excluded from the analysis making this group $n = 9$. All treatments were given at ZT8.5 with the exception of the OGTT. Animals were fed ad libitum except in pair-feeding studies and the OGTT. For Figs 6H–K and EV4F–I, body composition was determined weekly in live animals using an EchoMRI™ Body Composition Analyzer (EchoMRI™ Corp., Singapore). In all

experiments performed post-sacrifice, mice were euthanized and tissues collected 4 h after treatment.

## Voluntary cage running

To monitor voluntary exercise, sixteen HFD-fed mice were singly housed in cages equipped with running wheels and treated with vehicle or 120 mg/kg SH-BC-893 ($n = 8$). A magnet was affixed to each 240-mm wheel and a bicycle odometer (Sigma BC509, Sigma Sports, Chicago) used to count the number of wheel revolutions and time spent on running on the wheels. Distance run was calculated by the equation (#revolutions × running wheel circumference = distance). The wheels were cleaned and randomly re-assigned weekly to each cage to control for differences in wheel performance. Two animals in the SH-BC-893-treated group were euthanized due to gavage errors and were excluded from the analysis resulting in $n = 6$.

## Blood glucose measurements and oral glucose tolerance tests (OGTT)

Mice were fasted for 6 h prior to blood glucose testing at ZT10. When SH-BC-893 treatment was combined with an OGTT, mice were treated at ZT6. Once baseline fasting blood glucose was determined using a handheld blood glucose meter (Prodigy Diabetes Care, Charlotte, NC) and a drop of blood collected from a tail vein nick, mice were gavaged with an oral glucose solution (20% w/v in water, 2 g/kg body weight) and blood glucose measured in a drop of tail vein blood at 0, 15, 30, 60, and 120 min. The area under the curve was determined using GraphPad Prism software.

## Blood adipokine and insulin measurements

Blood was collected from the saphenous vein at ZT12.5, 4 h post-gavage with vehicle (water) or 120 mg/kg 893. Blood was allowed to clot at room temperature for 30 min and then centrifuged at 8,000 RPM at 4°C for 20 min. Hormones were measured using a mouse adiponectin ELISA kit (cat# 80569, Crystal Chem, Downers Grove, IL), mouse leptin ELISA kit (cat# 90030, Crystal Chem), or mouse insulin ELISA kit (cat# 90080, Crystal Chem) according to manufacturer's instructions. Serum was diluted 1:3 or 1:5 for leptin or 1:10,000 for adiponectin measurements in the diluent provided in the kit. The adiponectin:leptin ratio was calculated using [adiponectin] in µg/ml and [leptin] in ng/ml.

## RNA isolation and RT–qPCR analysis

RNA was extracted from flash-frozen eWAT or ARC tissues after a 24-h incubation in RNAlater™-ICE Frozen Tissue Transition Solution (AM7030, Thermo Fisher) at −20°C. Tissues were homogenized in 1 ml of TRI reagent (R2050-1-200, Zymo Research Corporation) using VWR® 200 homogenizer and RNA isolated using a Direct-zol RNA Purification Kit (R2051, Zymo Research Corporation). Quantity and integrity of RNA was monitored with a NanoDrop™ 2000c Spectrophotometer (Thermo Fisher Scientific). For cDNA synthesis, 1 µg of RNA was added to each reaction using iScript™ cDNA Synthesis Kit (Bio-Rad). Quantitative real-time PCR was carried out using a StepOnePlus™ Real-Time PCR System (Thermo Fisher Scientific).

For the analysis of *Socs3* and *Lepr*, the hypothalamus was dissected as described in (Seoane-Collazo & López, 2018). The relative gene expression was calculated by the comparative threshold cycle method and normalized to the expression of the housekeeping gene, *Rps16*. All RT–qPCR primers were taken from the Harvard Primer-Bank (https://pga.mgh.harvard.edu/primerbank/): *Socs3* (For: 5′-ATGGTCACCCACAGCAAGTTT-3′ and Rev: 5′-TCCAGTAGAATCC GCTCTCCT-3′), *Lepr* (For: 5′-TGGTCCCAGCAGCTATGGT and Rev: 5′-ACCCAGAGAAGTTAGCACTGT-3′), *Rps16* (For: 5′-CACTGCAAA CGGGGAAATGG-3′ and Rev: 5′-CACCAGCAAATCGCTCCTTG-3′), *Chop* (For: 5′-GTCCCTAGCTTGGCTGACAGA-3′ and Rev: 5′-TGGAG AGCGAGGGCTTTG-3′), *Grp78* (For: 5′-TCATCGGACGCACTTGG AA-3′ and Rev: 5′-CAACCACCTTGAATGGCAAGA-3′), *Xbp1s*: (For: 5′-GAGTCCGCAGCAGGTG-3′ and Rev: 5′-GTGTCAGAGTCCATG GGA-3′), and *Atf6* (For: 5′-GTCCAAAGCGAAGAGCTGTCTG-3′ and Rev: 5′-AGAGATGCCTCCTCTGATTGGC-3′).

## Indirect calorimetry

Metabolic parameters were measured using the Phenomaster system (TSE Systems Inc., Chesterfield, MO). The climate chamber was set to 21°C and 50% humidity with a 12-h:12-h light–dark cycle. Mice were maintained on the HFD or SD for 10–12 weeks prior to evaluation and then singly housed and acclimated for 48 h prior to data collection. Mice were evaluated in cohorts of 4 vehicle- and 4 SH-BC-893-treated mice using 8 metabolic cages. $VO_2$, $VCO_2$, and food intake were measured every 27 min. Respiratory exchange ratio (RER) was calculated using the formula $RER = VCO_2/VO_2$. Energy expenditure was calculated using the equation $EE = 1.44$ $(3.941 \times VO_2 + 1.106 \times VCO_2)$. For the pair-feeding study (Fig 6E), RER was monitored over 12 h in mice maintained on the HFD for 22 weeks (31 weeks of age); pair-fed mice were used after a 48-h wash-out period and provided with the average amount of food eaten in the 24 h after SH-BC-893 treatment (92.5 mg or 0.4 kcal). As is commonly observed, food intake was lower in metabolic cages than in communal, standard caging, but the same trends were observed in both forms of housing.

Data exclusion was as follows. Occasionally, uneaten food was found on the floor of the cage precluding use of the hopper sensor to accurately monitor food intake. In these instances, food intake data were censored for the prior 24-h period (day 3 for mouse 8 (HFD + SH-BC-893) and mouse 1 (HFD + vehicle) in Fig 6A and B). In rare cases, inadvertent pharyngeal administration of gavage material occurred. These mice were not euthanized, but food intake, calorimetry, and activity data from these animals were excluded from the analysis for 1 week after this event (mouse 2 (HFD + SH-BC-893) after the second treatment on day 3 and mouse 5 (HFD + SH-BC-893) after the first dose on day 1, Fig 6A–D).

## Home-cage feeding studies

For the studies shown in Figs 6O–R and EV4J and K, C57BL/6J mice maintained on a HFD for 9-10 weeks starting at 6 weeks of age were housed in groups of three and allowed to acclimate for 72 h before food intake was monitored. Food consumption was determined by monitoring the weight of food in the hopper. Initial food and body weight measurements were taken at ZT9 and final measurements were taken 24 h later to capture the active period where most

consumption occurred. Mice received vehicle or 120 mg/kg SH-BC-893 by gavage at ZT8.5.

## Lipidomic profiling

Lipids were extracted from liver and quadriceps tissue using a modified MTBE method (Matyash et al, 2008; Abbott et al, 2013). Briefly, 10 mg/ml of tissue was homogenized in ice-cold 150 mM ammonium acetate using a bead homogenizer (1.4 mm ceramic) kept below 4°C using liquid nitrogen vapor (Precellys 24 homogenizer with Cryolys cooling unit, Bertin Technologies, Montigny-le-Bretonneux, France). From this, 20 μl of homogenized tissues were added to glass vials containing MTBE and methanol (3:1 v/v, with 0.01% BHT), alongside 10 μl of an internal standard solution containing 10 μM each: phosphatidylcholine (PC) 17:0/17:0, phosphatidylethanolamine (PE) 17:0/17:0, phosphatidylserine (PS) 17:0/17:0, phosphatidylglycerol (PG) 17:0/17:0, lysophosphatidylcholine (LPC) 17:0, lysophosphatidylethanolamine (LPE) 14:0, ceramide (Cer) d18:1/17:0, dihydrosphingomyelin d18:0/12:0, diacylglycerol (DAG) 17:0/17:0, D5-triacylglycerol (TAG) 48:0, and cholesteryl ester (CE) 22:1. Samples were allowed to rotate at 4°C overnight prior to the addition of 1 volume of ice-cold 150 mM ammonium acetate. Samples were vortexed thoroughly prior to centrifugation (2,000× $g$, 5 min) to enable phase separation. The upper organic phase was removed to a new vial and dried under a stream of nitrogen with gentle heating (37°C). The dried lipids were reconstituted in chloroform:methanol:water (60:30:4.5 v/v/v) and kept at −20°C until analysis.

Extracted lipids were analyzed by liquid chromatography–mass spectrometry (LC-MS) using a Dionex Ultimate 3000 LC pump and Q Exactive Plus mass spectrometer equipped with a heated electrospray ionization (HESI) source (Thermo Fisher Scientific) (Hu et al, 2008; Bird et al, 2011; Turner et al, 2018). Lipids were separated on a Water ACQUITY C18 reverse-phase column (2.1 x 100 mm, 1.7-μm pore size, Waters Corp., Milford, MA) using a binary gradient, where mobile phase A consisted of acetonitrile:water (6:4 v/v) and B of isopropanol: acetonitrile (9:1 v/v). Both mobile phases A and B contained 10 mM ammonium formate and 0.1% formic acid, the flow rate was 0.26 ml/min, and the column oven was heated to 60°C. Source conditions were as follows: a spray voltage of 4.0 and 3.5 kV in positive and negative ion modes, respectively, capillary temperature of 290°C, S lens RF of 50, and auxiliary gas heater temperature of 250°C. Nitrogen was used as both source and collision gas, with sheath and auxiliary gas flow rate set at 20 and 5 (arbitrary units), respectively. Data were acquired in full scan/data-dependent MS2 mode (full-scan resolution 70,000 FWHM, max ion injection time 50 ms, scan range $m/z$ 200–1,500), with the 10 most abundant ions being subjected to collision-induced dissociation using an isolation window of 1.5 Da and a normalized stepped collision energy of 15/27 eV, with product ions detected at a resolution of 17,500. An exclusion list for background ions was developed using extraction blanks, and mass calibration was performed in both positive and negative ionization modes prior to analysis to ensure mass accuracy of 5 ppm in full-scan mode.

Lipids were analyzed using MS-DIAL (Tsugawa et al, 2015). Lipids were detected in both positive and negative ionization modes using a minimum peak height of $1 \times 10^4$ cps, a MS1 tolerance of 5 ppm and MS2 tolerance of 10 ppm, and a minimum identification

score of 50%. Identified peaks were aligned with a retention time tolerance of 0.5 min. Exported aligned data were background subtracted and quantified from internal standards using the statistical package R. One-way ANOVA with Tukey post hoc analysis was used to identify differences between groups with statistical significance set at an adjusted $P < 0.05$.

## Targeted metabolite quantification

C16:0 ceramide levels were quantified in cells (Fig 1I) using the method described in Kasumov et al (2010) with minor modifications. Cultured cells were washed twice in PBS and scraped into 250 μl of HPLC-grade water and flash-frozen until time of analysis. On the day of analysis, samples were thawed, and an aliquot was used for protein quantification. For C16:0 ceramide levels in mouse liver, 25 mg of tissue was homogenized in 1 ml of ice-cold PBS using a mechanical probe homogenizer (VWR, Radnor, PA), protein levels quantified, and 50 μl of the homogenate diluted with 150 μl HPLC-grade water for C16:0 ceramide analysis. Fifty ng of C17:0 ceramide prepared in ethanol (#22532, Cayman Chemical, Ann Arbor, MI) was added into 200 μl of the thawed cell suspension or liver homogenate as an internal standard to control for varying extraction efficiency; 750 μl of an ice-cold 1:2 chloroform/methanol mixture was then added. Samples were sonicated for 30 min and phase separation induced by the addition of 250 μl each of chloroform and HPLC-grade water. Samples were centrifuged at 4°C for 10 min and the lower lipid phase transferred to a clean tube. The remaining protein and aqueous layers were re-extracted with an additional 500 μl of chloroform. Lipid phases were combined and then dried under vacuum. Dried extract was reconstituted in 100% acetonitrile immediately before analysis. Samples were analyzed by ultra-performance liquid chromatography–tandem mass spectrometry (UPLC-MS/MS) using a Waters Micromass Quattro Premier XE equipped with a Waters ACQUITY BEH C4 column (Waters Corp.). Samples were resolved starting at 60% mobile phase A (10 mM ammonium acetate and 0.05% formic acid in water) to 98% mobile phase B (60:40 acetonitrile:isopropanol) over 3 min with a linear gradient, held at 98% B for 1 min, and then the column was equilibrated with 60% A for 1 min. The mass spectrometer was operated in positive ion mode with the following parameters: cone voltage 20 V, source temperature 125°C, desolvation temperature 400°C. Ion transition channels for MS/MS were 538 → 264 for C16:0 ceramide and 552 → 264 for C17:0 ceramide, both with a dwell time of 285 ms. Standard curves prepared from C16:0 ceramide (#860516, Avanti Polar Lipids, Alabaster, AL) dissolved in ethanol were used for quantitation and were linear from 4.1 nM – 1,000 nM, with an $R^2$ of 0.98 or greater.

## Cell culture

$p53^{flox/flox}$ MEFs were derived in house in 2015 from C57BL/6 mice using standard techniques and immortalized by transient expression of Cre recombinase and deletion of $p53$. MEFs were cultured and maintained in DMEM with 10% FBS and 1% penicillin–streptomycin. Stock solution of palmitic acid (ACROS Organics, cat# AC129702500) was prepared at 100 mM in ethanol. Palmitate (250 μM) was conjugated to 1% (w/v) fatty acid-free bovine serum albumin (Sigma, A8806) by incubation in DMEM at 37°C for 20 min before each experiment. For all immunofluorescence assays, 8,000

MEFs were seeded into 8-chamber slides (Cellvis, cat# C8-1.5H-N) 12–16 h before treatment. Cells were pre-treated with YM201636 (800 nM in DMSO), NAV2729 (12.5 μM in DMSO), SH-BC-893 (5 μM in water), myriocin (10 μM in methanol), fumonisin B1 (30 μM in DMSO), witherferin A (500 nM in DMSO), or celastrol (500 nM in DMSO) for 3 h or with FCCP (1 μM in DMSO) for 1 h followed by a 3-h treatment with BSA-conjugated palmitate or BSA alone. Where cells were treated with C16:0 ceramide (100 μM in ethanol) for 3 h, cells were pre-treated with SH-BC-893 for 1–3 h as indicated, mdivi-1 and M1 together for 1 or 24 h, leflunomide (50 μM in methanol) for 1 or 24 h, or with P110 (in water) for 1 h (10 μM) or for 12 h (1 μM). All cell lines were tested for *Mycoplasma* by PCR every 4–6 months using the VENOR GeM PCR Kit or using the protocol outlined in (Uphoff & Drexler, 2014). In addition, cells were evaluated for punctate DAPI staining that would be consistent with *Mycoplasma* infection every time microscopy was performed. None of the cells used in these experiments tested positive for *Mycoplasma* by either method.

## Plasmids and constructs

pEF6-Myc-GRP1-S155D/T240D was provided by Dr Victor W. Hsu (Harvard Medical School, Boston, MA, USA) and mCherry-TRPML1 plasmid was provided by Dr. Haoxing Xu (University of Michigan) and inserts sub-cloned into the doxycycline-inducible lentiviral plasmid pCW57.1 (Addgene Plasmid # 41393) via Gateway cloning. MEFs were transduced and selected with 3 μg of puromycin. Mitochondrial networks were evaluated 36-h post addition of 1 μg/ml doxycycline.

## Western blotting

To determine total DRP1 protein levels, 100,000 MEFs were seeded into a 6-well plate and 16 h later pre-treated for 3 h with vehicle or SH-BC-893 (5 μM) followed by a 3-h incubation with 1% BSA + ethanol or BSA-palmitate (250 μM). Cells were washed once with cold PBS and then lysed in cold RIPA buffer (140 mM NaCl, 10 mM Tris pH 8.0, 1% Triton X-100, 0.1% SDS, 1% sodium deoxycholate) with cOmplete™ protease inhibitor (Cat no. 11697498001, Millipore Sigma, St. Louis, MO) and phosSTOP™ phosphatase inhibitor (Cat no. 4906837001, Millipore Sigma). Samples were incubated on ice for 10 min and insoluble material removed by centrifugation (9,000 ×*g* for 10 min at 4°C). Protein content was quantified in the supernatant using the Pierce™ BCA Protein Assay Kit (Thermo Fisher Scientific, Waltham, MA). Equal amounts of protein were prepared in NuPAGE® LDS Sample Buffer (NP0007, Invitrogen) containing 50 mM DTT and heated at 70°C for 10 min. Proteins were resolved on a NuPAGE® 4–12% Bis-Tris protein gel (NP0336, Invitrogen, Carlsbad, CA) and subsequently transferred to a nitrocellulose membrane. Membranes were blocked in 5% BSA in TBST for 1 h and then probed with primary antibodies overnight at 4°C. Antibodies used were rabbit-anti-DRP1 at 1:1,000 (#8570, Cell Signaling Technology) and mouse anti-tubulin at 1:10,000 (T8328, Millipore Sigma, St. Louis, MO). Blots were then washed 3X in TBST and incubated in 800CW-conjugated goat anti-rabbit (#926-32211, Li-COR, Lincoln, NB) and 680LT-conjugated goat anti-mouse (#925-68020, Li-COR) secondary antibodies at 1:10,000 in 5% BSA in TBST for 1 h. Blots were washed then imaged using a Li-COR

Odyssey CLx instrument. Band intensity was quantified using Image Studio Lite V5.2 software (Li-COR).

## Microscopy

MEFs were washed twice with PBS and fixed with 4% paraformaldehyde for 10 min at RT. Cells were permeabilized with 0.3% Triton X-100 in blocking buffer containing 10% fetal bovine serum for 20 min at 37°C followed by overnight incubation with mouse anti-citrate synthase (sc-390693, Santa Cruz Biotechnology; dilution, 1:200) or rabbit-anti-DRP1 at 1:100 (#8570, Cell Signaling Technology) at 4°C. Cells were then washed twice with PBS and incubated with AlexaFluor 488 goat anti-mouse (A28175, Invitrogen) or Alexa-Fluor 594 donkey anti-rabbit (A32754, Invitrogen) secondary antibodies at RT followed by 5 min incubation with 1 μg/ml DAPI and 2 washes in PBS. For mitochondrial membrane potential measurements MEFs were pre-treated with SH-BC-893 (5 μM in water) for 3 h and followed by a 12 h treatment with BSA-conjugated palmitate or BSA alone. Cells were incubated with 100 nM tetramethylrhodamine ethyl ester, perchlorate (TMRE, catalog #82720, Thermo Fisher Scientific) and 200 nM MitoTracker™ Green FM (Catalog # M7514, Thermo Fisher Scientific) for 15 min in DMEM followed by two washes with PBS and imaged in DMEM without phenol red. For ROS measurements, MEFs were pre-treated with SH-BC-893 (5 μM) for 3 h followed by a 12-h treatment with C2-ceramide (catalog #62510, Cayman Chemical, 50 μM in DMSO). Cells were incubated with 5 μM MitoSOX™ Red Mitochondrial Superoxide Indicator (catalog #M36008, Thermo Fisher Scientific) for 30 min in PBS followed by two PBS washes and cells were imaged in DMEM without phenol red. For NAD(P)H autofluorescence studies, mice were gavaged with vehicle or 120 mg/kg SH-BC-893 at ZT8.5 (4–10.5 h before sacrifice between ZT12.5 and ZT18). Post-sacrifice, the liver or epididymal fat pad (eWAT) was excised, washed 3× with PBS, placed in DMEM supplemented with 10% FBS and 1% penicillin–streptomycin, and immediately imaged with 740 nm excitation using a Mai Tai two-photon laser and 450 ± 50 nm detectors. Fluorescence microscopy was performed on a Zeiss LSM 780 confocal using a 63× oil objective with a 1.7 numerical aperture (NA) or on a Nikon TE2000-S inverted epifluorescence microscope with a 100× oil objective (1.3 NA) and a Photometrics CoolSNAP ES2 monochrome CCD camera. For evaluating mitochondrial function and ROS in the eWAT, fat pads were cut into 8 equal pieces and placed in PBS containing 200 nM MTG and MitoSOX (5 μM) or TMRE (100 nM), and tissues washed twice with PBS before evaluation in DMEM without phenol red. All confocal images are 16-bit images from 8 to 15 Z-stacks with 0.5-micron steps or 16-bit single slices. At least 6-12 non-overlapping fields of view were obtained. Confocal images were obtained using Zeiss Zen 2.3 image acquisition software. To analyze the co-localization between DRP1/citrate synthase signals, Mander's overlap coefficient was calculated using the JACOP co-localization plug-in of ImageJ v.1.52e (NIH) after background subtraction on a per field basis; 40 cells were analyzed from 2 biological replicates. For H&E staining, livers were fixed in formalin, dehydrated in ethanol, and processed by the Experimental Tissue Research pathology core facility at UCI and evaluated on a Nikon Ti2-F inverted epifluorescence microscope equipped with a DS-Fi3 color camera. Five non-overlapping fields were acquired from 3 different liver sections obtained from 3 mice per group (SD, HFD, or HFD + 120 mg/kg SH-BC-893) and

representative images shown. For imaging of brain mitochondria, mice were perfused transcardially with PBS followed by 4% paraformaldehyde immediately after euthanasia. Whole brains were removed, incubated in 4% paraformaldehyde at 4°C for 24 h, and then transferred to a 30% sucrose solution in 0.1 M PBS for storage. On the day of sectioning, brains were equilibrated in OCT for 5-10 min prior to freezing in the cryostat chamber (CM 1850-3-1; Leica Microsystems) at −20°C in cryoboats. Free-floating coronal sections of 30 µm were generated. To evaluate the arcuate nucleus of the hypothalamus, the coordinates −0.5 to −2.4 mm were determined using a mouse brain atlas (Franklin & Paxinos, 2001). Sections were blocked and permeabilized with 5% normal horse serum in 0.5% Triton X-100 at 4°C for 24 h, incubated for 48 h at 4°C with citrate synthase primary antibody (1:100), washed, incubated with Alexa Fluor 488-conjugated secondary antibody (1:200), and counter-stained with DAPI before mounting in Vectashield. No fluorescence was observed when primary antibodies were omitted. Images from 6 non-overlapping fields in 2 different sections were evaluated from each of 3 mice per group (HFD + vehicle or HFD + 120 mg/kg SH-BC-893) using a Zeiss LSM 780 confocal microscope and a 63× oil objective. The image of the full coronal brain section was taken a 4× magnification with an Axiocam camera mounted on an Axioskop-2 epifluorescence microscope (Zeiss) using Zen lite software; a composite image was stitched together using Adobe Photoshop version 9.0.

### Morphometric quantification of mitochondrial networks

Schematics describing the quantitative analysis of mitochondrial networks are provided as Fig EV1 (in vitro) and Fig EV3 (in vivo). Analysis was performed using ImageJ software as described in (Chaudhry *et al*, 2019). Briefly, maximum projections from Z-stacks were pre-processed to remove background, manually thresholded as necessary to accurately capture mitochondria, and binarized images evaluated using the analyze particles tool (roundness = $4 \times$ area/$\pi \times$ width and aspect ratio = width/height) or skeletonized and analyzed using the analyze skeleton 2D/3D function (branch length). Cell boundaries were manually delimited using the brightfield channel. For in vivo samples, noise was reduced with the despeckle function. Branch length was not calculated for in vivo samples as they were minimally branched. In vitro analysis was performed on 40 cells (20 cells from each of the 2 biological replicates) from 6 to 10 non-overlapping fields of view. In each cell, 100–500 objects were evaluated and averaged; average values from 40 individual cells were used to generate averages for each condition. Analysis of liver and brain mitochondria was performed on a per field basis using 6–12 non-overlapping fields collected for each animal. NAD(P)H images from the liver were subjected to the Tikhonov-Miller deconvolution algorithm (regularization parameter of 5.0; 50 iteration and background subtraction of 50 pixel rolling ball radius) using ImageJ via Deconvolutionlab2 open-source plug-in.

### Statistical analysis

In bar graphs or scatterplots, mean ± SEM or mean ± SD, respectively, is presented unless otherwise indicated in the legends. All experimental data are from ≥ 3 independent biological replicates except where otherwise indicated. Statistical analysis was performed

### The paper explained

#### Problem

The rates of obesity and its lethal comorbidities are high worldwide and continue to rise. Therapies that reverse obesity and its metabolic sequelae are an unmet clinical need. Consuming a high-fat, Western diet leads to obesity because it triggers excessive mitochondrial fission downstream from increased ceramide generation. Consistent with this, genetically engineered mouse models have revealed that fragmenting mitochondrial networks in metabolic tissues is sufficient to produce the leptin and insulin resistance that drives obesity and metabolic dysfunction. Despite this connection, small molecules targeting the mitochondrial fission and fusion machinery offered only minimal benefits in diet-induced obesity models so far.

#### Results

A novel approach, modulating mitochondrial dynamics by disrupting endolysosomal trafficking, proved more effective than previously tested agents targeting mitochondrial proteins. The orally bioavailable, water-soluble synthetic sphingolipid SH-BC-893 prevented palmitate and ceramide-induced fragmentation of mitochondrial networks by disrupting ARF6- and PIKfyve-dependent endolysosomal trafficking. Consistent with its robust in vitro activity and favorable pharmacologic properties, SH-BC-893 normalized mitochondrial morphology in the liver, brain, and white adipose tissue of mice with diet-induced obesity within 4 h of oral administration. In keeping with its ability to improve mitochondrial function and correct ER stress, SH-BC-893 acutely reversed hyperleptinemia, restored leptin sensitivity, and reduced food intake. SH-BC-893 was well tolerated and effective as an interventional agent, normalizing body weight and correcting metabolic dysfunction despite continued consumption of a high-fat diet. The benefits of SH-BC-893 were additive with voluntary exercise, suggesting that it could complement other weight loss strategies.

#### Impact

Our findings suggest a novel approach to treating obesity. If SH-BC-893 proves similarly effective and safe in human patients, it could protect from the lethal sequelae of obesity both indirectly, by triggering weight loss, and directly, as cardiovascular disease, liver dysfunction, and cancer are also linked to unbalanced mitochondrial fission. Given its rapid and robust effects in multiple tissues, the effects of SH-BC-893 therapy in other conditions linked to excessive mitochondrial fission (e.g., neurodegenerative diseases, addiction, and depression) merit investigation.

using GraphPad Prism software except for lipid profiling when the statistical package R was used. Corrections for multiple comparisons were made as indicated in the legends and adjusted *P*-values reported: ns, not significant, $P \geq 0.05$; *$P < 0.05$; **$P < 0.01$; ***$P < 0.001$; key comparisons are shown in the figures. For all figure panels, exact *P*-values for all comparisons are provided in the Appendix.

## Data availability

This study includes no data deposited in external repositories.

**Expanded View** for this article is available online.

## Acknowledgements

The authors would like to thank the following present and former UCI colleagues for generously sharing equipment, reagents, and/or providing

technical advice: Raquel Chamorro and Bruce Blumberg (EchoMRI), Katrina Waymire (mouse work), Carl Cottman (running wheels), Felix Grun (LC-MS/MS), Mei Kong and Eric Hanse (adipocyte culture), Wenqi Wang (DS-Fi3 color camera), Adela Syeed and Michelle Digman (in vivo imaging), Sima Chokr (brain sectioning), and Brendan Finicle (TRPML1 and GRP1-DD expression plasmids). We would also like to thank Matheus Viana and Susanne Rafelski (Allen Institute for Cell Science, Seattle WA) for providing advice regarding quantification of mitochondrial networks. This project was supported by NIH R01 CA254360 and a Proof of Product grant from Beall Applied Innovation to ALE; the National Center for Research Resources and the National Center for Advancing Translational Sciences, National Institutes of Health, through Grant UL1 TR001414; a National Health and Medical Research Council of Australia project grant (1126135) to NT; and by grants from the National Institutes of Health, the Concern Foundation for Cancer Research, and the V Foundation for Cancer Research to SM. ES and VJ were supported by the National Cancer Institute of the National Institutes of Health under award number T32CA009054, ES was also supported by Graduate Assistance in Areas of National Need (GAANN) grant P200A120207. AV was supported by a Hitachi-Nomura postdoctoral fellowship. The authors also wish to acknowledge the support of the Chao Family Comprehensive Cancer Center Optical Biology Center and Experimental Tissue Resource shared resources, supported by the National Cancer Institute of the National Institutes of Health under award number P30CA062203. The content is solely the responsibility of the authors and does not necessarily represent the official views of the National Institutes of Health. Subsidized access to the UNSW Bioanalytical Mass Spectrometry Facility is also gratefully acknowledged. SH and BC acknowledge financial support from NSERC Canada.

## Author contributions

ES, VJ, AV, SM, NT, and ALE formulated the overarching research goals. All authors contributed to the development or design of the methodology. ES, VJ, SEH, KHE, BMH, GM, MOG, and AV conducted the research and ES, VJ, SEH, and ALE analyzed the data and prepared the figures. BC synthesized SH-BC-893. SM, SH, KSC, AGF, NT, and ALE supervised the research and acquired financial support leading to this publication. VJ, ES, and ALE prepared the original draft of the manuscript, and all authors revised and approved the final version.

## Conflict of interest

ALE and SH are founders of Siege Pharmaceuticals which is developing SH-BC-893 for use in cancer and other diseases. Other authors declare no conflict of interest.

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
