## [Review Process File · EMBO Molecular Medicine]

Drug-like sphingolipid SH-BC-893 opposes ceramide-induced mitochondrial fission and corrects diet-induced obesity

Vaishali Jayashankar, Elizabeth Selwan, Sarah Hancock, Amandine Verlande, Maggie Goodson, Kazumi Eckenstein, Giedre Milinkeviciute, Brianna Hoover, Bin Chen, Angela Fleischman, Karina Cramer, Stephen Hanessian, Selma Masri, Nigel Turner, and Aimee Edinger

DOI: [10.15252/emmm.202013086](https://doi.org/10.15252/emmm.202013086)

Corresponding author(s): *Aimee Edinger (aedinger@uci.edu)*

Review Timeline:

Submission Date:	9th Jul 20
Editorial Decision:	14th Aug 20
Revision Received:	30th Apr 21
Editorial Decision:	27th May 21
Revision Received:	2nd Jun 21
Accepted:	9th Jun 21

Editor: *Lise Roth*

Transaction Report:

14th Aug 2020

Dear Dr. Edinger,

Thank you for submitting your work to EMBO Molecular Medicine. We have now heard back from the three referees who agreed to evaluate your manuscript. As you will see below, the reviewers find that the question addressed by the study is of interest, however, they remain unconvinced that some of the major conclusions are sufficiently supported by the data.

In particular, they agree that a better characterization/understanding of the food intake/satiety effects is needed, and also think that the current data should be improved from a technical standpoint.

Addressing the reviewers concerns in full, either experimentally or in writing, will be necessary for further considering the manuscript in our journal. Still, revising the manuscript according to the referees' recommendations appears to require a lot of additional work and experimentation. Thus, and given the current pandemic situation, we are ready to extend the deadline to 6 months with the understanding that acceptance of the manuscript would entail a second round of review. EMBO Molecular Medicine encourages a single round of revision only and therefore, acceptance or rejection of the manuscript will depend on the completeness of your responses included in the next, final version of the manuscript. For this reason, and to save you from any frustrations in the end, I would strongly advise against returning an incomplete revision.

When submitting your revised manuscript, please carefully review the instructions that follow below. Failure to include requested items will delay the evaluation of your revision:

- 1) A .docx formatted version of the manuscript text (including legends for main figures, EV figures and tables). Please make sure that the changes are highlighted to be clearly visible.
- 2) Individual production quality figure files as .eps, .tif, .jpg (one file per figure).
- 3) A .docx formatted letter INCLUDING the reviewers' reports and your detailed point-by-point responses to their comments. As part of the EMBO Press transparent editorial process, the point-by-point response is part of the Review Process File (RPF), which will be published alongside your paper.
- 4) A complete author checklist, which you can download from our author guidelines (<https://www.embopress.org/page/journal/17574684/authorguide#submissionofrevisions>). Please insert information in the checklist that is also reflected in the manuscript. The completed author checklist will also be part of the RPF.
- 5) Please note that all corresponding authors are required to supply an ORCID ID for their name upon submission of a revised manuscript.
- 6) Before submitting your revision, primary datasets produced in this study need to be deposited in an appropriate public database (see <https://www.embopress.org/page/journal/17574684/authorguide#dataavailability>).

Please remember to provide a reviewer password if the datasets are not yet public. The accession numbers and database should be listed in a formal "Data Availability " section (placed after Materials & Method). Please note that the Data Availability Section is restricted to new primary data that are part of this study.

7) We would also encourage you to include the source data for figure panels that show essential data. Numerical data should be provided as individual .xls or .csv files (including a tab describing the data). For blots or microscopy, uncropped images should be submitted (using a zip archive if multiple images need to be supplied for one panel). Additional information on source data and instruction on how to label the files are available at .

8) Our journal encourages inclusion of *data citations in the reference list* to directly cite datasets that were re-used and obtained from public databases. Data citations in the article text are distinct from normal bibliographical citations and should directly link to the database records from which the data can be accessed. In the main text, data citations are formatted as follows: "Data ref: Smith et al, 2001" or "Data ref: NCBI Sequence Read Archive PRJNA342805, 2017". In the Reference list, data citations must be labeled with "[DATASET]". A data reference must provide the database name, accession number/identifiers and a resolvable link to the landing page from which the data can be accessed at the end of the reference. Further instructions are available at .

9) We replaced Supplementary Information with Expanded View (EV) Figures and Tables that are collapsible/expandable online. A maximum of 5 EV Figures can be typeset. EV Figures should be cited as 'Figure EV1, Figure EV2' etc... in the text and their respective legends should be included in the main text after the legends of regular figures.

- Additional Tables/Datasets should be labeled and referred to as Table EV1, Dataset EV1, etc. Legends have to be provided in a separate tab in case of .xls files. Alternatively, the legend can be supplied as a separate text file (README) and zipped together with the Table/Dataset file. See detailed instructions here: .

10) The paper explained: EMBO Molecular Medicine articles are accompanied by a summary of the articles to emphasize the major findings in the paper and their medical implications for the non-specialist reader. Please provide a draft summary of your article highlighting

11) For more information: There is space at the end of each article to list relevant web links for

further consultation by our readers. Could you identify some relevant ones and provide such information as well? Some examples are patient associations, relevant databases, OMIM/proteins/genes links, author's websites, etc...

12) Every published paper now includes a 'Synopsis' to further enhance discoverability. Synopses are displayed on the journal webpage and are freely accessible to all readers. They include a short stand first (maximum of 300 characters, including space) as well as 2-5 one-sentences bullet points that summarizes the paper. Please write the bullet points to summarize the key NEW findings. They should be designed to be complementary to the abstract - i.e. not repeat the same text. We encourage inclusion of key acronyms and quantitative information (maximum of 30 words / bullet point). Please use the passive voice. Please attach these in a separate file or send them by email, we will incorporate them accordingly.

Please also suggest a striking image or visual abstract to illustrate your article. If you do please provide a png file 550 px-wide x 400-px high.

13) As part of the EMBO Publications transparent editorial process initiative (see our Editorial at <http://embomolmed.embopress.org/content/2/9/329>), EMBO Molecular Medicine will publish online a Review Process File (RPF) to accompany accepted manuscripts.

In the event of acceptance, this file will be published in conjunction with your paper and will include the anonymous referee reports, your point-by-point response and all pertinent correspondence relating to the manuscript. Let us know whether you agree with the publication of the RPF and as here, if you want to remove or not any figures from it prior to publication.

I look forward to receiving your revised manuscript.

Yours sincerely,

Lise Roth

Lise Roth, PhD
Editor
EMBO Molecular Medicine

To submit your manuscript, please follow this link:

Photos 400-800 DPI

*Additional important information regarding figures and illustrations can be found at <http://bit.ly/EMBOPressFigurePreparationGuideline>

***** Reviewer's comments *****

Referee #1 (Remarks for Author):

Selwan et al. demonstrate that the drug-like sphingolipid SH-BC-893 corrects high-fat diet-induced obesity in mice. They show that this interesting therapeutic effect is linked to reversing mitochondrial fragmentation in the liver and hypothalamus. The weight loss effect is driven by reduced food intake -presumably to sensitization to leptin. Although there are some questions regarding acute versus chronic modes of action of SH-BC-893 treatment on hypothalamic neurons and food intake, the effects on fat mass and blood sugar are remarkable. The technical quality is very high, except for a relative low number of mice in their cohorts and some open questions in the ob/ob experiments. The manuscript is very well written. My major and minor points of constructive criticism are listed below:

Major:

1. The major question at the end is whether the drug acts on exclusively on hypothalamic regulation of energy intake and how is this related to leptin levels and leptin sensitivity. First, as the story is constructed around the effect on food intake and leptin, I think the authors should include plasma leptin levels in the DIO model and if possible, provide data on how these change during the course of the SH-BC-893 treatment. In the end, while the data appear clear in that SH-BC-893 acutely inhibits feeding in both DIO and ob/ob mice, only on DIO mice this results on weight loss when applied chronically. To my opinion, this conclusion is based on a potentially underpowered study on ob/ob. The treatment period in ob/ob was relatively short and the cumulative food intake "looks different" (even though, there are no errors bars in Fig 7h so hard to tell what is going on). So even if there is no reduction in body weight with reduced food intake, I think this set of experiments would benefit from a more reliable number of mice (a more robust reproduction) and also by providing more data on energy metabolism here for ob/ob as was done for DIO mice (Fig. 6, needless to ask what about insulin levels, liver phenotype etc in ob/ob). Right now, to me it is relatively unclear, which effects of SH-BC-893 treatment are related to leptin and which are not.

Minor:

2. Body weight is a major denominator of metabolic health - in the catabolic state, upon weight loss, most pathological hallmarks in diet-induced obesity (especially in mice that have been only fed a couple of weeks) are reversed, including non-alcoholic fatty liver disease. To my opinion, the reduced steatosis upon SH-BC-893 treatment just follows the body weight loss and the finding does not allow concluding that there is an independent effect of the drug on NASH development.

3. In addition to plasma glucose, plasma insulin levels should be provided during the course of the treatment experiments.

4. When the authors compared SH-BC-893 to other agents, they compared a treatment of 5 μ M SH-BC-893 to a treatment of 1 μ M P110 and concluded that the treatment SH-BC-893 is superior. Is there a reason why the authors did not use the same concentration for both drugs? Would SH-BC-893 still be superior when compared to the same concentration of P110? Please elaborate.

Referee #2 (Comments on Novelty/Model System for Author):

To address a potential CNS-specific effect of 893 (on hypothalamic POMC neurons), stereotaxic brain injection should be conducted to limit the action of the compound to the brain.

Referee #2 (Remarks for Author):

(1) The authors demonstrate that compound SH-BC-893 sensitizes leptin-induced anorectic effect in HFD-fed mice that are leptin-resistant. Leptin-mediated central effects include suppression of feeding, increase in energy expenditure as well as enhancement of the regulation of glucose homeostasis and reduction of hepatic steatosis. To assess the significance of leptin-sensitizing effect of SH-BC-893 in hypothalamic neurons, central administration of the compound by stereotaxic injection may be considered. At least, there should be some discussion as a technical limitation in the manuscript.

(2) Can the authors provide more mechanistic insight on the effects of mitochondrial fission on mitochondrial function?

(3) The authors suggest that compound 893 inhibits ceramide-induced recruitment of DRP1 to mitochondria. Since the compound does not reduce C16:0 ceramide synthesis, what is the underlying molecular mechanism in the blockade of DRP1 mitochondrial translocation?

(4) Did compound 893 treatment affect HFD-induced ER stress? Mfn2 (mitochondrial fusion machinery) has been implicated in unfolded protein responses (UPR) triggered by ER stress.

Referee #3 (Remarks for Author):

The manuscript from Selwan and colleagues describes the action of a drug-like sphingolipid to treat obesity by reversing ceramide-induced mitochondrial fragmentation. Changes in mitochondrial dynamics have been studied extensively in obesity but the lack of efficient pharmacological drugs directly targeting mitochondrial fragmentation made it difficult to assess this target as a valid

therapeutic strategy. The authors used the sphingolipid SH-BC-893 to inhibit ceramides production and test whether this lipid can improve metabolism in response to palmitate in cells and in obese mice models. Selwan and colleagues showed that unlike their primary hypothesis, 893 blocks mitochondrial fragmentation downstream of ceramide production as 893 failed to reduced ceramide levels in MEF cells. This is associated with more Drp1 recruitment and is more efficient to block mitochondrial fragmentation than other known drugs. Mitochondrial morphology is preserved in mice fed HFD and treated with SH-BC-893 compared to WT. Obese mice treated with SH-BC-893 lose fat mass, body weight, and have better glucose homeostasis due to lower food intake. The authors provide also further data showing that SH-BC-893 reduces food intake without reducing body weight in ob/ob mice suggesting that some effects of SH-BC-893 are leptin dependent. Although the premise of the study and the sphingolipid analog, SH-BC-893, is interesting, many interpretations of the data are beyond the results presented here. Besides, in the present version, all the metabolic phenotypes observed in HF fed mice are most likely due to lower food consumption. Many of the models used were not appropriate to answers the author's hypothesis. Although the results are promising, the present manuscript does not shed more light on the mechanism by which the SH-BC-893 regulates food intake, mitochondrial fragmentation, and acts as an anti-obesity therapy.

Addressing the following concerns will improve this manuscript:

Major comments

1. The authors concluded that SH-BC-893 improved glucose homeostasis, insulin sensitivity, and increase fat oxidation while the main effect of SH-BC-893 is the reduction in food intake and most likely appetite. Based on the current data, SH-BC-893 does not directly act on glucose homeostasis, insulin sensitivity, and fat oxidation as SH-BC-893 induces these effects because of reducing the energy intake and body weight. Unless the authors do a Pair-fed study showing that these effects are independent of food intake, the authors should modify their conclusions which reduces the impact of this sphingolipid analog on metabolism as these effects are expected from an anti-obesity therapy. Also, the reduce ceramide levels in the liver can just be explained by lower food intake. As the effect on food intake is the main mechanism, that result should be shown in figure 4. If the authors want to evaluate insulin sensitivity and glucose homeostasis in vivo independently of food intake, hyperinsulinemic-euglycemic clamps and GTTs should be performed in pair-fed mice.
2. As the main action of SH-BC-893 is on food intake, the authors should focus their experimental approach to explain the mechanisms by which this sphingolipid act on the neurocircuitry involved in appetite sensing and regulation. Identifying whether this lipid act on AGRP and POMC neurons is essential to better understand the role of SH-BC-893 and ceramides on food intake in obesity.
3. The authors suggested leptin as the mechanism by which SH-BC-893 affects food intake. The ob/ob model used to answer this question has a defect in the leptin gene which prevents its expression and its secretion. Although the idea is interesting, the data does not support the hypothesis that the leptin is driving the effect mediated by SH-BC-893 as the SH-BC-893 still reduced food intake although leptin is not produced. Also, the authors compared the wrong group in the data performed in fig. 7a and b. These panels do not show that SH-BC-893 actions are dependent on leptin as the two green columns are not significantly different. It only shows that the leptin cannot reduce food intake and body weight more than the SH-BC-893 already did. To conclude that the effects of SH-BC-893 are leptin-dependent, the authors should have used a leptin receptor-deficient model such as the db/db mice. Also, the fact that mitochondrial morphology is different (more elongated) than the HFD mice suggest that it is not a good model to assess the effect of ceramides on mitochondrial morphology (presence of an adaptative mechanism in ob/ob or the absence of high-fat diet?). Also, if the author's hypothesis is right, why looking at mitochondrial morphology in the resected liver and not in the white adipose tissue which produces leptin?
4. The imaging technic used by the authors to assess mitochondrial morphology is correct but

further analysis is required to be more convincing. First, all the imaging was done in fixed cells which can alter mitochondrial morphology and structure. Experiments using live cells should be performed and in other cell types of interest than MEFs cells such as neurons or primary white adipocyte. Second, the imaging analysis to assess mitochondrial morphology in tissues was performed with NADH/NADPH autofluorescence in resected organs which will change by itself mitochondrial morphology over time. Electron microscopy should be performed for resected organs. Otherwise, immunofluorescence on histology slides including co-staining to identify the cell type of interest would be better. There is also a lot of background using citrate synthase immunofluorescence and NADH/NADPH autofluorescence in tissues in fig.3 which make it hard to see mitochondria.

5. Metabolic studies should be performed in more than 4 mice only... Why in fig.4 there were 8-10 mice used for fat mass analysis but only 4 mice for GTT in fig.5?

6. Mitochondrial fragmentation is not always a synonym of mitochondrial dysfunction. Mitochondrial fragmentation is required to adapt to changes in fuel preference, notably fat oxidation. However, fasting is associated with both mitochondrial elongation and an increase in fat oxidation as well. As changes in mitochondrial morphology is not a clear indicator of fuel preference, the authors should measure mitochondrial fuel preference from glucose/pyruvate vs lipid oxidation on mitochondrial respiratory capacity in isolated mitochondria and/or cells treated with SH-BC-893. This will help understand the action of the SH-BC-893 on mitochondrial function and not just morphology.

7. The 3T3-L1 and 2DG uptake experiments to assess insulin sensitivity should be done in the presence of ceramides to determine if SH-BC-893 can revert the insulin resistance induced by ceramides. In the blot Fig.5 C, how did you quantify pS473 Akt from the wells without insulin? No bands are visible.

8. Page 16 "). However, SH-BC-893-treated ob/ob mice still consumed more food than the treated wild type, HFD-fed controls suggesting a role for leptin in the anorexigenic actions of SH-BC-89". The fold change in food intake induced by the SH-BC-893 is the same. So, the SH-BC-893 did the same job in ob/ob mice than in the HF fed-mice.

9. As SH-BC-893 acts on mitochondrial morphology downstream of ceramide synthesis, the authors suggest that the main mechanism of SH-BC-893 is by preventing mitochondrial fragmentation in obesity which affects leptin and insulin sensitivity. However, to prove that, experiments using white adipocyte-specific (Adiponectin-Cre)-mfn2 ko mice on HFD treated with SH-BC-893 should be performed.

10. Are Drp1 and Drp1 phosphorylation important for the actions of SH-BC-893 to prevent mitochondrial fragmentation? Can SH-BC-893 prevent mitochondrial fragmentation in cells independently of Drp1? Drp1 recruitment experiment should be performed in the cell type of interest such as neurons and white adipocyte.

11. In the text, the authors referred a lot to the effect of the SH-BC-893 or the ceramides in the liver while they are proposing a mechanism linking white adipose tissue-leptin and the brain. The manuscript should focus on this axis and cell types unless the authors think that SH-BC-893 also affects glucose metabolism in the liver independently of lower food intake and body weight in vivo which more experiments are required to answer this question.

Minor comments:

- Figure 6a is confusing as mice were treated with SH-BC-893 only on days 1 and 3. Add the meaning of the stars directly on the figure.
- All the exercise and fig.2 data should be in supplemental as it does not bring more information about the mechanism.
- Is the DRP1 fluorescence in fig.1 J, similar between 893 and control conditions? Fluorescence intensity seems lower in 893 treated cells.
- Scale bar is missing in fig. 2I
- The text in the results sections doesn't match the order of the panels in the figures which make the manuscript difficult to read

We would like to thank the reviewers for their insightful comments. Responding to their concerns has resulted in substantial improvements to the manuscript.

**** Reviewer's comments ****

Referee #1 (Remarks for Author):

Selwan et al. demonstrate that the drug-like sphingolipid SH-BC-893 corrects high-fat diet-induced obesity in mice. They show that this interesting therapeutic effect is linked to reversing mitochondrial fragmentation in the liver and hypothalamus. The weight loss effect is driven by reduced food intake -presumably to sensitization to leptin. Although there are some questions regarding acute versus chronic modes of action of SH-BC-893 treatment on hypothalamic neurons and food intake, the effects on fat mass and blood sugar are remarkable. The technical quality is very high, except for a relative low number of mice in their cohorts and some open questions in the ob/ob experiments. The manuscript is very well written. My major and minor points of constructive criticism are listed below:

Major:

1. The major question at the end is whether the drug acts on exclusively on hypothalamic regulation of energy intake and how is this related to leptin levels and leptin sensitivity. First, as the story is constructed around the effect on food intake and leptin, I think the authors should include plasma leptin levels in the DIO model and if possible, provide data on how these change during the course of the SH-BC-893 treatment. In the end, while the data appear clear in that SH-BC-893 acutely inhibits feeding in both DIO and ob/ob mice, only on DIO mice this results on weight loss when applied chronically. To my opinion, this conclusion is based on a potentially underpowered study on ob/ob. The treatment period in ob/ob was relatively short and the cumulative food intake "looks different" (even though, there are no errors bars in Fig 7h so hard to tell what is going on). So even if there is no reduction in body weight with reduced food intake, I think this set of experiments would benefit from a more reliable number of mice (a more robust reproduction) and also by providing more data on energy metabolism here for ob/ob as was done for DIO mice (Fig. 6, needless to ask what about insulin levels, liver phenotype etc in ob/ob). Right now, to me it is relatively unclear, which effects of SH-BC-893 treatment are related to leptin and which are not.

The revised manuscript now demonstrates that SH-BC-893 (893) has profound effects on WAT mitochondria that lead to reduced plasma leptin levels offering an additional mechanism for leptin sensitization that likely complements changes in hypothalamic mitochondrial morphology (Figs. 3-5 and EV5B). Both leptin and adiponectin are measured in HFD-fed mice dosed once or repeatedly (Fig. 5) as suggested. We demonstrate that reduced food intake is not sufficient to explain altered adipokine levels (Fig. 5A,B) consistent with the model that improved mitochondrial function in WAT (Fig. 4) reduces leptin secretion (Fig. EV5B). Both lowering plasma leptin (Fig. 5) and opposing mitochondrial fission in the hypothalamus (Fig. 3G-I) will sensitize to leptin and reduce food intake.

Regarding the previously underpowered study in ob/ob mice, we performed a follow up study with a larger group size (n=10). We obtained results consistent with those we reported in the original submission. Male B6 ob/ob mice treated with 893 did not lose weight or show improvements in hepatic steatosis or glucose handling (OGTT) after 3 wk of treatment. However, we came to the same conclusion as Reviewer 3 – the ob/ob model is not ideal to assess the contribution of leptin sensitization to 893's metabolic effects. As we reported in the prior submission, hepatic mitochondria in ob/ob mice are fused despite elevated liver ceramides. New experiments revealed that the mitochondrial phenotype in the WAT and brain of ob/ob mice was also distinct from HFD-fed mice. In the WAT of ob/ob mice with obesity, MitoSOX staining intensity was not increased (ROS was elevated in the WAT of HFD-fed mice with obesity, Fig. 4H-J). In addition, there was extremely low basal NAD(P)H signal in ob/ob eWAT that was unaffected by SH-BC-893 (NAD(P)H autofluorescence increased in both chow- and HFD-fed wild type mice, Fig. 4B,C). Finally, mitochondrial morphology in the ARC of ob/ob mice was not altered relative to wild type, chow-fed controls (HFD-fed mice with obesity exhibited a decreased aspect ratio and an increased roundness, Fig. 3G-I). Given these differences between the mitochondrial phenotype in wild type and ob/ob mice with obesity, it is unclear whether ob/ob mice fail to respond to 893 because they lack leptin or because mitochondrial fragmentation is not driving obesity in the ob/ob

background. We have therefore removed data in the ob/ob model from the paper and will prepare a separate manuscript describing the differences in the mitochondrial phenotypes of weight-matched HFD-fed and ob/ob animals.

To conclude, we feel that the new data on adipokines (Fig. 5), mechanistic insights (Fig. 1A-D and N-Q), and data mitochondrial function assays (Figs. 2 and 4) added in response to Reviewer comments have significantly improved the manuscript and support our revised model focused on the WAT-brain axis (Fig. EV5B).

Minor:

2. Body weight is a major denominator of metabolic health - in the catabolic state, upon weight loss, most pathological hallmarks in diet-induced obesity (especially in mice that have been only fed a couple of weeks) are reversed, including non-alcoholic fatty liver disease. To my opinion, the reduced steatosis upon SH-BC-893 treatment just follows the body weight loss and the finding does not allow concluding that there is an independent effect of the drug on NASH development.

We agree with the reviewer that weight loss would be a key mediator of any beneficial effects of 893 on fatty liver disease. Given that SH-BC-893 also increases circulating adiponectin levels (Fig. 5A, D, and J) and that blocking mitochondrial fission may reduce hepatic steatosis independent of weight loss (Galloway *et al*, 2014), 893 may also have beneficial effects on the liver that are independent of body weight reduction. We now clarify in the text that additional work, particularly in models consuming fructose, will be required to determine the potential value of 893 in addressing NAFLD/NASH.

3. In addition to plasma glucose, plasma insulin levels should be provided during the course of the treatment experiments.

We now show that 893 reduces plasma insulin (Fig. 7H,I). Because pair feeding phenocopies the effect of 893, it is likely that insulin levels are reduced secondary to the reduction in food intake. However, as mentioned in the Discussion, it would be worthwhile to evaluate the effects of 893 on beta cell mitochondria in future studies.

4. When the authors compared SH-BC-893 to other agents, they compared a treatment of 5 μ M SH-BC-893 to a treatment of 1 μ M P110 and concluded that the treatment SH-BC-893 is superior. Is there a reason why the authors did not use the same concentration for both drugs? Would SH-BC-893 still be superior when compared to the same concentration of P110? Please elaborate.

The 1 μ M P110 dose was selected based on literature reports demonstrating that this is an effective dose. We now present data with P110 at 10 μ M (Fig. EV2I-L); 1 and 10 μ M P110 produce equivalent outcomes.

Referee #2 (Comments on Novelty/Model System for Author):

To address a potential CNS-specific effect of 893 (on hypothalamic POMC neurons), stereotaxic brain injection should be conducted to limit the action of the compound to the brain.

With the discovery that SH-BC-893 has dramatic effects on WAT mitochondria and reduces circulating leptin (Figs. 4 and 5), the model has been revised and is no longer dependent on direct, central effects of the compound for reduced food intake (Fig. EV5B). We discuss the potential value of stereotaxic brain injection in the Discussion as suggested by the reviewer (see below). However, the effects of 893 in multiple brain areas (Fig. 3F-L) and the potential for 893 to spread within the brain or enter the circulation (893 crosses the BBB) during the time required for analysis of food intake might complicate interpretation of these studies.

Referee #2 (Remarks for Author):

(1) The authors demonstrate that compound SH-BC-893 sensitizes leptin-induced anorectic effect in HFD-fed mice that are leptin-resistant. Leptin-mediated central effects include suppression of feeding, increase in energy expenditure as well as enhancement of the regulation of glucose homeostasis and reduction of hepatic steatosis. To assess the

significance of leptin-sensitizing effect of SH-BC-893 in hypothalamic neurons, central administration of the compound by stereotaxic injection may be considered. At least, there should be some discussion as a technical limitation in the manuscript.

We thank the reviewer for this suggestion and now discuss the value of performing icv injection in future studies in the Discussion (see comment above).

(2) Can the authors provide more mechanistic insight on the effects of mitochondrial fission on mitochondrial function?

We have added mitochondrial function data from MEFs (Fig. 2) and WAT (Figs. 4). This data shows that SH-BC-893 restores the mitochondrial membrane potential, reduces ROS, and reduces ER stress. In WAT, SH-BC-893 also increases NAD(P)H levels.

(3) The authors suggest that compound 893 inhibits ceramide-induced recruitment of DRP1 to mitochondria. Since the compound does not reduce C16:0 ceramide synthesis, what is the underlying molecular mechanism in the blockade of DRP1 mitochondrial translocation?

Consistent with literature reports now cited in the text (Wong *et al*, 2018; Peng *et al*, 2020; Li *et al*, 2017; Abuarab *et al*, 2017; Farmer *et al*, 2017), 893's established effects on endolysosomal trafficking limit DRP1 recruitment (Fig. 1A-D and N-Q). These "necessary and sufficient" experiments where the PIKfyve and ARF6 pathways are manipulated in the context of our published data demonstrating how 893 alters endolysosomal trafficking (Kim *et al*, 2016; Finicle *et al*, 2018) strongly support the model in Fig. EV5B.

(4) Did compound 893 treatment affect HFD-induced ER stress? Mfn2 (mitochondrial fusion machinery) has been implicated in unfolded protein responses (UPR) triggered by ER stress.

Thank you for this suggestion, we now show that SH-BC-893 reduces palmitate- (Fig. 2) and HFD-induced (Fig. 4) ER stress.

Referee #3 (Remarks for Author):

The manuscript from Selwan and colleagues describes the action of a drug-like sphingolipid to treat obesity by reversing ceramide-induced mitochondrial fragmentation. Changes in mitochondrial dynamics have been studied extensively in obesity but the lack of efficient pharmacological drugs directly targeting mitochondrial fragmentation made it difficult to assess this target as a valid therapeutic strategy. The authors used the sphingolipid SH-BC-893 to inhibit ceramides production and test whether this lipid can improve metabolism in response to palmitate in cells and in obese mice models. Selwan and colleagues showed that unlike their primary hypothesis, 893 blocks mitochondrial fragmentation downstream of ceramide production as 893 failed to reduced ceramide levels in MEF cells. This is associated with more Drp1 recruitment and is more efficient to block mitochondrial fragmentation than other known drugs. Mitochondrial morphology is preserved in mice fed HFD and treated with SH-BC-893 compared to WT. Obese mice treated with SH-BC-893 lose fat mass, body weight, and have better glucose homeostasis due to lower food intake. The authors provide also further data showing that SH-BC-893 reduces food intake without reducing body weight in ob/ob mice suggesting that some effects of SH-BC-893 are leptin dependent. Although the premise of the study and the sphingolipid analog, SH-BC-893, is interesting, many interpretations of the data are beyond the results presented here. Besides, in the present version, all the metabolic phenotypes observed in HF fed mice are most likely due to lower food consumption. Many of the models used were not appropriate to answers the author's hypothesis. Although the results are promising, the present manuscript does not shed more light on the mechanism by which the SH-BC-893 regulates food intake, mitochondrial fragmentation, and acts as an anti-obesity therapy.

The revised manuscript clearly shows that 893 corrects disruptions in circulating adipokine levels in mice with HFD-induced obesity upstream of its effects on food intake (Fig. 5A,B) and provides a molecular mechanism of action (Fig. 1A-D and N-Q and (Wong *et al*, 2018; Farmer *et al*, 2017; Peng *et al*, 2020; Abuarab *et al*, 2017; Li *et al*, 2017)). As pointed

out by Reviewer 3 and highlighted in Fig. EV2, 893's profound and rapid effects on mitochondrial morphology both in vitro (Figs. 1 and 2) and in vivo (Figs. 3 and 4) make it superior to previously described compounds used to target mitochondrial dynamics. SH-BC-893's acute (4 h after administration) effects on plasma leptin and adiponectin levels (Fig. 5) and mitochondrial form/function in liver, brain, and WAT (Figs. 3 and 4) are striking - to our knowledge, there are no reports in the literature of a compound with these exciting, disease-relevant activities. This study not only highlights a novel therapeutic strategy to oppose mitochondrial fission that will stimulate new research directions, it provides an important proof of principle that therapeutics that modulate mitochondrial fission in multiple tissues in vivo could be well tolerated. Finally, SH-BC-893's oral bioavailability, favorable PK properties, and apparently low toxicity make it a viable pre-clinical candidate. All of these features will make this manuscript of significant interest to readers of a translationally-focused journal like *EMBO Molecular Medicine* and to the obesity and metabolic disease communities in general.

Addressing the following concerns will improve this manuscript:

Major comments

1. The authors concluded that SH-BC-893 improved glucose homeostasis, insulin sensitivity, and increase fat oxidation while the main effect of SH-BC-893 is the reduction in food intake and most likely appetite. Based on the current data, SH-BC-893 does not directly act on glucose homeostasis, insulin sensitivity, and fat oxidation as SH-BC-893 induces these effects because of reducing the energy intake and body weight. Unless the authors do a Pair-fed study showing that these effects are independent of food intake, the authors should modify their conclusions which reduces the impact of this sphingolipid analog on metabolism as these effects are expected from an anti-obesity therapy. Also, the reduce ceramide levels in the liver can just be explained by lower food intake. As the effect on food intake is the main mechanism, that result should be shown in figure 4. If the authors want to evaluate insulin sensitivity and glucose homeostasis in vivo independently of food intake, hyperinsulinemic-euglycemic clamps and GTTs should be performed in pair-fed mice.

We apologize if the prior manuscript was confusing on this point, but we did not wish to propose that SH-BC-893 directly affects glucose homeostasis or fat oxidation – we showed that the reduced RER can be accounted for by reduced food intake (retained in the current manuscript as Fig. 6E). We agree that the metabolic benefits of 893 are likely to stem largely from reduced food intake and weight loss (e.g. Fig. 7I). However, a new pair-feeding study demonstrates that leptin and adiponectin are changed independent of reduced food intake (Fig. 5A-C). Increased circulating adiponectin (Fig. 5A,D, and J) may increase sensitivity to insulin and reduce ceramide levels (adiponectin receptors possess ceramidase activity) independent of food intake. We anticipate that the new data, reorganization of the paper, and the addition of a model (Fig. EV5B) will make the hypotheses easier to follow.

2. As the main action of SH-BC-893 is on food intake, the authors should focus their experimental approach to explain the mechanisms by which this sphingolipid act on the neurocircuitry involved in appetite sensing and regulation. Identifying whether this lipid act on AGRP and POMC neurons is essential to better understand the role of SH-BC-893 and ceramides on food intake in obesity.

In the context of paradigm-shifting work from the Scherer group (Zhao *et al*, 2019), our findings that SH-BC-893 reduces circulating leptin (Fig. 5B, E, and K) and reduces *Socs3* mRNA and increases *LepR* mRNA (Fig. 5G,H) in the hypothalamus offers convincing evidence for an indirect mechanism for leptin sensitization. Central effects also likely contribute. In chow-fed mice where leptin levels are not significantly decreased (Fig. 5K), 893 still reduces food intake (Figs. 6O,P and EV4J,K). While this finding is consistent published results in lean mice where DRP1 is knocked out in POMC neurons (Santoro *et al*, 2017) and suggests that central actions of 893 also reduce food intake, additional work will be required to separate the direct and indirect mechanisms by which 893 could restore leptin sensitivity in mice with obesity.

3. The authors suggested leptin as the mechanism by which SH-BC-893 affects food intake. The ob/ob model used to answer this question has a defect in the leptin gene which prevents its expression and its secretion. Although the idea is interesting, the data does not support the hypothesis that the leptin is driving the effect mediated by SH-BC-893 as the

SH-BC-893 still reduced food intake although leptin is not produced. Also, the authors compared the wrong group in the data performed in fig. 7a and b. These panels do not show that SH-BC-893 actions are dependent on leptin as the two green columns are not significantly different. It only shows that the leptin cannot reduce food intake and body weight more than the SH-BC-893 already did. To conclude that the effects of SH-BC-893 are leptin-dependent, the authors should have used a leptin receptor-deficient model such as the db/db mice. Also, the fact that mitochondrial morphology is different (more elongated) than the HFD mice suggest that it is not a good model to assess the effect of ceramides on mitochondrial morphology (presence of an adaptive mechanism in ob/ob or the absence of high-fat diet?). Also, if the author's hypothesis is right, why looking at mitochondrial morphology in the resected liver and not in the white adipose tissue which produces leptin?

We are very grateful for the Reviewer's suggestion to evaluate mitochondria in WAT. Collecting this data (Fig. 4) helped to refine our model and significantly improve the manuscript.

The old Fig. 7A/B was no longer supported by a strong rationale once we realized that *reduced* leptin levels were likely contributing to decreased food intake; this figure was removed. After collecting additional data in ob/ob mice (see comments to Reviewer 1), we agree with Reviewer 3 that this model is not useful to determine whether SH-BC-893's metabolic benefits in mice with HFD-induced obesity depend on leptin. Literature suggests that db/db mice have profound mitochondrial defects and would suffer from similar limitations (Choo *et al*, 2006).

4. The imaging technic used by the authors to assess mitochondrial morphology is correct but further analysis is required to be more convincing. First, all the imaging was done in fixed cells which can alter mitochondrial morphology and structure. Experiments using live cells should be performed and in other cell types of interest than MEFs cells such as neurons or primary white adipocyte. Second, the imaging analysis to assess mitochondrial morphology in tissues was performed with NADH/NADPH autofluorescence in resected organs which will change by itself mitochondrial morphology over time. Electron microscopy should be performed for resected organs. Otherwise, immunofluorescence on histology slides including co-staining to identify the cell type of interest would be better. There is also a lot of background using citrate synthase immunofluorescence and NADH/NADPH autofluorescence in tissues in fig.3 which make it hard to see mitochondria.

We now evaluate mitochondria in both live and fixed cells using additional, complementary techniques: NAD(P)H autofluorescence, citrate synthase staining, MitoTracker Green, TMRE, and MitoSOX. Primary adipocytes were examined (Fig. 4). Our data in freshly resected eWAT and liver was collected immediately after sacrifice; mice were euthanized in pairs to ensure tissues could be imaged quickly. All in vivo data is robust and reproducible (Figs. 3 and 4); we would argue that this data from intact tissues may be more physiologically relevant than data collected in cultured cells. We also confirm in Fig. EV3B that NAD(P)H and MitoTracker Green signals completely overlap. References are now provided to support our statement that our in vivo images are consistent with published studies using mice expressing mitochondrially-localized fluorescent proteins. Finally, liver NAD(P)H images were subjected to deconvolution and we improved citrate synthase staining by working with experts in brain imaging (new co-authors Giedre Milinkeviciute and Karina Cramer).

5. Metabolic studies should be performed in more than 4 mice only... Why in fig.4 there were 8-10 mice used for fat mass analysis but only 4 mice for GTT in fig.5?

In the OGTT, the effect size is large and the variability is low; the experiments were appropriately powered with n=4. We have clarified that these animals were selected at random. Resolution of hyperinsulinemia (Fig. 7H) is also consistent with the conclusions drawn from the OGTT.

6. Mitochondrial fragmentation is not always a synonym of mitochondrial dysfunction. Mitochondrial fragmentation is required to adapt to changes in fuel preference, notably fat oxidation. However, fasting is associated with both mitochondrial elongation and an increase in fat oxidation as well. As changes in mitochondrial morphology is not a clear indicator of fuel preference, the authors should measure mitochondrial fuel preference from glucose/pyruvate vs lipid

oxidation on mitochondrial respiratory capacity in isolated mitochondria and/or cells treated with SH-BC-893. This will help understand the action of the SH-BC-893 on mitochondrial function and not just morphology.

We apologize if the earlier manuscript was confusing, but we did not intend to propose that SH-BC-893 alters mitochondrial fuel preference. The reduction in the RER can be completely accounted for by the reduction in food intake (Fig. 6E); the decreased RER reflects the catabolism of stored fat. Rewiring of metabolism by 893 is not required to explain our results.

7. The 3T3-L1 and 2DG uptake experiments to assess insulin sensitivity should be done in the presence of ceramides to determine if SH-BC-893 can revert the insulin resistance induced by ceramides. In the blot Fig.5 C, how did you quantify pS473 Akt from the wells without insulin? No bands are visible.

Given the reorganization of the manuscript to accommodate the new model and mechanistic data, we have replaced the insulin signaling panels looking at AKT with a reference to our published phosphoproteomics paper that demonstrates that, unlike ceramide or other PP2A agonists, SH-BC-893 does not reduce the phosphorylation of AKT or its substrates (Kubiniok *et al*, 2019). The original data was sound - a low signal was measured in the ROI drawn for quantification of pAKT in the absence of insulin and was normalized to total protein as per standard practice in the field. This panel was eliminated because it did not contribute much to the new manuscript.

8. Page 16 "). However, SH-BC-893-treated ob/ob mice still consumed more food than the treated wild type, HFD-fed controls suggesting a role for leptin in the anorexigenic actions of SH-BC-89". The fold change in food intake induced by the SH-BC-893 is the same. So, the SH-BC-893 did the same job in ob/ob mice than in the HF fed-mice.

As discussed above, we have removed data with the ob/ob model from the manuscript consistent with the Reviewer's suggestion.

9. As SH-BC-893 acts on mitochondrial morphology downstream of ceramide synthesis, the authors suggest that the main mechanism of SH-BC-893 is by preventing mitochondrial fragmentation in obesity which affects leptin and insulin sensitivity. However, to prove that, experiments using white adipocyte-specific (Adiponectin-Cre)-mfn2 ko mice on HFD treated with SH-BC-893 should be performed.

The elegant published work with the (Adiponectin-Cre)-Mfn2 KO mice (Mancini *et al*, 2019) greatly informed the development of our experimental plan and new model (Fig. EV5B). Importantly, knocking out Mfn2 in individual tissues is expected to only partially undermine 893's effects given the parallel actions alluded to above. We respectfully suggest that this work in genetically-engineered mice is beyond the scope of the present manuscript. We now highlight in the Discussion that studies in Mfn2 KO mice are an important future direction.

10. Are Drp1 and Drp1 phosphorylation important for the actions of SH-BC-893 to prevent mitochondrial fragmentation? Can SH-BC-893 prevent mitochondrial fragmentation in cells independently of Drp1? Drp1 recruitment experiment should be performed in the cell type of interest such as neurons and white adipocyte.

We now demonstrate that SH-BC-893's established inhibitory effects on ARF6 and PIKfyve (Kim *et al*, 2016; Finicle *et al*, 2018) are necessary and sufficient to account for its ability to oppose ceramide-induced mitochondrial fission (Fig. 1A-D and N-Q). We also point out that Drp1 phosphorylation was not altered in our published, unbiased phosphoproteomics study with SH-BC-893 (Kubiniok *et al*, 2019).

11. In the text, the authors referred a lot to the effect of the SH-BC-893 or the ceramides in the liver while they are proposing a mechanism linking white adipose tissue-leptin and the brain. The manuscript should focus on this axis and cell types unless the authors think that SH-BC-893 also affects glucose metabolism in the liver independently of lower food intake and body weight in vivo which more experiments are required to answer this question.

We thank the Reviewer for the very helpful suggestion to focus on the brain-WAT axis. This suggestion led to key insights that now form the basis of our revised model (Fig. EV5B).

Minor comments:

* Figure 6a is confusing as mice were treated with SH-BC-893 only on days 1 and 3. Add the meaning of the stars directly on the figure.

We now use a legend to indicate 893 dosing rather than an *.

* All the exercise and fig.2 data should be in supplemental as it does not bring more information about the mechanism.

We have retained some of the exercise data in Fig. 6L-N as we feel it is translationally relevant and therefore important to EMM's readers. SH-BC-893 was initially described as an anti-cancer agent, and whether it is toxic is a critical question likely to be in readers' minds. Moreover, new agents are generally tested in clinical trials in combination with lifestyle interventions (Adler, 2021); it is encouraging that combining 893 with exercise produces additional benefits.

* Is the DRP1 fluorescence in fig.1 J, similar between 893 and control conditions? Fluorescence intensity seems lower in 893 treated cells.

We include quantitative western blot data (n=3) to address this point – Drp1 levels are not altered by 893. It is common that cytoplasmic proteins are difficult to detect by IF until they are recruited to membranes and therefore more concentrated. For example, this same phenotype occurs when monitoring LC3 localization by IF when autophagy is induced.

* Scale bar is missing in fig. 2I

We have confirmed that all microscopy data includes scale bars.

* The text in the results sections doesn't match the order of the panels in the figures which make the manuscript difficult to read

Figures are discussed in the order presented.

Bibliography

- Abuarab N, Munsey TS, Jiang L-H, Li J & Sivaprasadarao A (2017) High glucose-induced ROS activates TRPM2 to trigger lysosomal membrane permeabilization and Zn²⁺-mediated mitochondrial fission. *Sci. Signal.* **10**:
- Adler AI (2021) The STEP 1 trial for weight loss: a step change in treating obesity? *Nat. Med.* **27**: 589–590
- Choo HJ, Kim JH, Kwon OB, Lee CS, Mun JY, Han SS, Yoon YS, Yoon G, Choi KM & Ko YG (2006) Mitochondria are impaired in the adipocytes of type 2 diabetic mice. *Diabetologia* **49**: 784–791
- Farmer T, Reinecke JB, Xie S, Bahl K, Naslavsky N & Caplan S (2017) Control of mitochondrial homeostasis by endocytic regulatory proteins. *J. Cell Sci.* **130**: 2359–2370
- Finicle BT, Ramirez MU, Liu G, Selwan EM, McCracken AN, Yu J, Joo Y, Nguyen J, Ou K, Roy SG, Mendoza VD, Corrales DV & Edinger AL (2018) Sphingolipids inhibit endosomal recycling of nutrient transporters by inactivating ARF6. *J. Cell Sci.* **131**:
- Galloway CA, Lee H, Brookes PS & Yoon Y (2014) Decreasing mitochondrial fission alleviates hepatic steatosis in a murine model of nonalcoholic fatty liver disease. *Am. J. Physiol. Gastrointest. Liver Physiol.* **307**: G632–41
- Kim SM, Roy SG, Chen B, Nguyen TM, McMonigle RJ, McCracken AN, Zhang Y, Kofuji S, Hou J, Selwan E, Finicle BT, Nguyen TT, Ravi A, Ramirez MU, Wiher T, Guenther GG, Kono M, Sasaki AT, Weisman LS, Potma EO, *et al* (2016) Targeting cancer metabolism by simultaneously disrupting parallel nutrient access pathways. *J. Clin. Invest.* **126**: 4088–4102

- Kubiniok P, Finicle BT, Piffaretti F, McCracken AN, Perryman M, Hanessian S, Edinger AL & Thibault P (2019) Dynamic Phosphoproteomics Uncovers Signaling Pathways Modulated by Anti-oncogenic Sphingolipid Analogs. *Mol. Cell Proteomics* **18**: 408–422
- Li F, Munsey TS & Sivaprasadarao A (2017) TRPM2-mediated rise in mitochondrial Zn²⁺ promotes palmitate-induced mitochondrial fission and pancreatic β -cell death in rodents. *Cell Death Differ.* **24**: 1999–2012
- Mancini G, Pirruccio K, Yang X, Blüher M, Rodeheffer M & Horvath TL (2019) Mitofusin 2 in mature adipocytes controls adiposity and body weight. *Cell Rep.* **26**: 2849–2858.e4
- Peng W, Wong YC & Krainc D (2020) Mitochondria-lysosome contacts regulate mitochondrial Ca²⁺ dynamics via lysosomal TRPML1. *Proc. Natl. Acad. Sci. USA* **117**: 19266–19275
- Santoro A, Campolo M, Liu C, Sesaki H, Meli R, Liu Z-W, Kim JD & Diano S (2017) DRP1 suppresses leptin and glucose sensing of POMC neurons. *Cell Metab.* **25**: 647–660
- Wong YC, Ysselstein D & Krainc D (2018) Mitochondria-lysosome contacts regulate mitochondrial fission via RAB7 GTP hydrolysis. *Nature* **554**: 382–386
- Zhao S, Zhu Y, Schultz RD, Li N, He Z, Zhang Z, Caron A, Zhu Q, Sun K, Xiong W, Deng H, Sun J, Deng Y, Kim M, Lee CE, Gordillo R, Liu T, Odle AK, Childs GV, Zhang N, *et al* (2019) Partial leptin reduction as an insulin sensitization and weight loss strategy. *Cell Metab.* **30**: 706–719.e6

27th May 2021

Dear Dr. Edinger,

Thank you for the submission of your revised manuscript to EMBO Molecular Medicine, and please accept my apologies for the delay in getting back to you, as we were waiting for the report from one referee.

We have now received the enclosed reports from referees #1 and #2. We have not heard back from referee #3, but referee #1 carefully looked at your responses to referee #3's concerns and was satisfied. I am therefore pleased to inform you that we will be able to accept your manuscript once the following editorial points will be addressed:

1/ Referees' comments:

Please address the minor concerns from referee #2.

2/ Main manuscript text:

- Please answer/correct the changes suggested by our data editors in the main manuscript file (in track changes mode). This file will be sent to you in the next couple of days. Please use this file for any further modification.
- Please remove the draft summary.
- Please place the Material and Methods section after the Discussion and before the Acknowledgements.
- Material and methods:
 - o Mice experiments: when appropriate, please indicate the age of the mice.
 - o Cells: please indicate whether cells were tested for mycoplasma contamination.
- Data Availability section: Before submitting your revision, primary datasets produced in this study need to be deposited in an appropriate public database, and the accession numbers and database should be listed in a formal "Data Availability" section (placed after Materials & Method). Please note that the Data Availability Section is restricted to new primary data that are part of this study. If no new datasets were generated, please indicate: "This study includes no data deposited in external repositories"
- The authors Maggie O. Goodson, Kazumi Eckenstein, Giedre Milinkeviciute, Brianna M. Hoover, Bin Chen, Angela G. Fleischman, Karina S. Cramer have not been entered in the submission system, please adjust accordingly. Please also make sure the Author Contributions section is complete.
- Please replace "Competing interest statement" by "Conflict of interest".
- Please reformat the references to have 10 authors listed before et al.

3/ Figures and Appendix:

- We do not accept .ppt format for figures, please upload individual production quality figure files as .eps, .tif, .jpg (one file per figure).
- Statistics: please indicate in the legends or in the figures the exact n= and exact p= values (including for non-significant p values, ns), along with the statistical test used. Some people found that to keep the figures clear, providing a supplemental table with all exact p-values was preferable. You are welcome to do this if you want to.
- Appendix: please convert the appendix to PDF and add a Table of Content.

4/ We would also encourage you to include the source data for figure panels that show essential

data. Numerical data should be provided as individual .xls or .csv files (including a tab describing the data). For blots or microscopy, uncropped images should be submitted (using a zip archive if multiple images need to be supplied for one panel). Additional information on source data and instruction on how to label the files are available at

5/ Checklist:

Section F/19: if no new dataset was generated, please indicate "This study includes no data deposited in external repositories"

6/ Please provide "The paper explained" section: EMBO Molecular Medicine articles are accompanied by a summary of the articles to emphasize the major findings in the paper and their medical implications for the non-specialist reader. Please provide a summary of your article highlighting

7/ Thank you for providing a synopsis text. Could you please modify it slightly to fit our style and format? The short stand-first should be 300 characters maximum (including space). The bullet points (2 to 5, 30 words maximum) should summarize the paper, and when possible written in the passive voice.

Please upload your synopsis image separately as a png/jpeg/tiff file (550x400px).

8/ As part of the EMBO Publications transparent editorial process initiative (see our Editorial at <http://embomolmed.embopress.org/content/2/9/329>), EMBO Molecular Medicine will publish online a Review Process File (RPF) to accompany accepted manuscripts.

This file will be published in conjunction with your paper and will include the anonymous referee reports, your point-by-point response and all pertinent correspondence relating to the manuscript. Let us know whether you agree with the publication of the RPF.

I look forward to receiving your revised manuscript.

Yours sincerely,

Lise Roth

Lise Roth, PhD
Editor
EMBO Molecular Medicine

To submit your manuscript, please follow this link:

Photos 400-800 DPI

*Additional important information regarding figures and illustrations can be found at <https://bit.ly/EMBOPressFigurePreparationGuideline>

The system will prompt you to fill in your funding and payment information. This will allow Wiley to send you a quote for the article processing charge (APC) in case of acceptance. This quote takes into account any reduction or fee waivers that you may be eligible for. Authors do not need to pay any fees before their manuscript is accepted and transferred to our publisher.

***** Reviewer's comments *****

Referee #1 (Remarks for Author):

Thank you for addressing all these points.

Referee #2 (Comments on Novelty/Model System for Author):

After extensive revision, all these parameters have become even better.

Referee #2 (Remarks for Author):

The authors have fully addressed my concerns raised on the initial version of the manuscript. I have several more minor concerns regarding the proposed model in Fig. 5B:

(1) Hypothalamic neuronal inflammation should not be left out as a potential mechanism underlying leptin resistance. Both oxidative stress and ER stress may result in activation of inflammation mediators such as NF-kappa B and JNK. The authors should briefly discuss inflammation.

(2) In the model, increase in leptin contributes to the increase in appetite, which is incorrect because leptin has potent anorectic effect. I believe the authors are saying hyperleptinemia, through promoting leptin resistance, contributes to the increase in appetite. Thus, "the increase in leptin" should be replaced as "hyperleptinemia".

The authors have addressed all minor editorial requests.

9th Jun 2021

Dear Aimee,

Thank you for sending the revised files.

I am very pleased to inform you that your manuscript is accepted for publication and is now being sent to our publisher to be included in the next available issue of EMBO Molecular Medicine.

Congratulations on your interesting work!

With my best wishes,

Lise

Lise Roth, Ph.D
Scientific Editor
EMBO Molecular Medicine

Follow us on Twitter @EmboMolMed
Sign up for eTOCs at embopress.org/alertsfeeds

*** ** IMPORTANT INFORMATION ** **

SPEED OF PUBLICATION

The journal aims for rapid publication of papers, using using the advance online publication "Early View" to expedite the process: A properly copy-edited and formatted version will be published as "Early View" after the proofs have been corrected. Please help the Editors and publisher avoid delays by providing e-mail address(es), telephone and fax numbers at which author(s) can be contacted.

Should you be planning a Press Release on your article, please get in contact with embomolmed@wiley.com as early as possible, in order to coordinate publication and release dates.

LICENSE AND PAYMENT:

All articles published in EMBO Molecular Medicine are fully open access: immediately and freely available to read, download and share.

EMBO Molecular Medicine charges an article processing charge (APC) to cover the publication costs. You, as the corresponding author for this manuscript, should have already received a quote with the article processing fee separately. Please let us know in case this quote has not been received.

Once your article is at Wiley for editorial production you will receive an email from Wiley's Author Services system, which will ask you to log in and will present you with the publication license form for completion. Within the same system the publication fee can be paid by credit card, an invoice, pro forma invoice or purchase order can be requested.

Payment of the publication charge and the signed Open Access Agreement form must be received before the article can be published online.

PROOFS

You will receive the proofs by e-mail approximately 2 weeks after all relevant files have been sent to our Production Office. Please return them within 48 hours and if there should be any problems, please contact the production office at embopressproduction@wiley.com.

Please inform us if there is likely to be any difficulty in reaching you at the above address at that time. Failure to meet our deadlines may result in a delay of publication.

All further communications concerning your paper proofs should quote reference number EMM-2020-13086-V3 and be directed to the production office at embopressproduction@wiley.com.

Thank you,

Lise Roth, Ph.D
Scientific Editor
EMBO Molecular Medicine

Corresponding Author Name: Aimee L. Edinger

Manuscript Number: EMM-2020-13086-V2